# Metabolic cardiomyopathies: untangling clinical heterogeneity with human stem-cell derived models

Adriana S Passadouro[1,2,3,4], Berith M Balfoort [iD] [1,5,6,7], Mirjam Langeveld[4,7], Clara D M van Karnebeek[5,6,8,9], Jolanda van der Velden[2,3], Riekelt H Houtkooper [iD] [1,3,4,6 ✉] & Signe Mosegaard [iD] [1,3,4,6 ✉]

## Abstract

**Inherited metabolic diseases are rare monogenic conditions that disrupt biochemical pathways, affecting energy production and homeostasis, often leading to damaging metabolite accumulation. These disorders are clinically heterogeneous and can impact all organs, including the heart. Metabolic cardiomyopathies present with varying severity and unpredictable prognosis, complicating patient care. Pre-clinical research aims to model these cardiomyopathies to understand their pathophysiological mechanisms and develop personalised treatments. Animal models have provided insights into cardiac pathology and treatment, but species differences limit data translation. Human induced pluripotent stem cells (hiPSC) offer a valuable tool for establishing disease models using reprogrammed somatic cells from patients and healthy donors, differentiated into disease-relevant cell types. Cardiomyocytes generated in significant numbers are crucial for investigating cardiac mechanisms and assessing patient-specific drug responses. This review summarises literature on metabolic cardiomyopathies, focusing on long-chain fatty acid oxidation disorders and Barth syndrome. We highlight cardiac readouts from various models and discuss the potential of hiPSC technologies as clinically relevant disease models.**

**Keywords** Metabolic Cardiomyopathy; Inherited Metabolic Diseases (IMD); Pre-clinical Models; Personalized Medicine
**Subject Categories** Cardiovascular System; Metabolism; Stem Cells & Regenerative Medicine

## Introduction

Cardiomyopathies are highly complex and heterogeneous disorders and many genetic and environmental factors contribute to the pathogenesis and progression of cardiac electrical or mechanical dysfunction. Cardiac metabolism has emerged as a key factor underlying cardiomyopathy onset and progression, given the strong connection between metabolic remodelling and defective contractility in pathological conditions. The 2023 guidelines from the European Society of Cardiology distinguish seven types of genetic cardiomyopathies, which include metabolic cardiomyopathies and inherited structural cardiomyopathies (Arbelo et al, 2023). Inherited metabolic disorders (IMDs) are rare monogenic conditions in which defects in specific enzymes or transporters disrupt biochemical pathways, compromising energy production and homeostasis, often accompanied by the accumulation of damaging metabolites. IMDs can cause metabolic cardiomyopathies; however, adaptive or maladaptive metabolic abnormalities can occur secondary to the variant-mediated cellular alterations in inherited structural cardiomyopathies such as hypertrophic (HCM) and dilated (DCM) cardiomyopathies (Fig. 1) (Ramaccini et al, 2021; van der Velden et al, 2018; Nollet et al, 2023).

HCM has a prevalence of 1:200, and it is mostly caused by pathogenic variants affecting sarcomeres (van der Velden et al, 2018; Sacchetto et al, 2019). These genetic defects translate into pathological hallmarks such as diastolic dysfunction, hypertrophy, fibrosis, and disarray (van der Velden et al, 2018; Chou and Chin, 2021). At the cardiomyocyte level, mutant sarcomeres have been shown to utilise more adenosine triphosphate (ATP) for force development (Witjas-Paalberends et al, 2014), and cardiac tissues of patients with obstructive HCM show disorganised mitochondria with reduced function, and perturbed cardiac metabolism (Nollet et al, 2023; Schuldt et al, 2021; Nollet et al, 2024; Ranjbarvaziri et al, 2021). DCM has a prevalence of 1:250–400, and patients present with ventricular dilation, arrhythmias and contractile dysfunction (Weintraub et al, 2017; Reichart et al, 2019). DCM can be caused by both non-genetic and genetic factors, with truncating titin (TTN) variants being the most prevalent genetic cause. These genetic defects have been linked to metabolic compensatory mechanisms such as significant transcriptional upregulation of oxidative phosphorylation (OXPHOS) (Verdonschot et al, 2018).

Cardiomyocytes represent a third of the cells in the heart, and 30% of their volume is occupied by mitochondria, which are responsible for coordinating metabolite generation, stress responses

[1]Laboratory Genetic Metabolic Diseases, Amsterdam UMC, Location University of Amsterdam, Amsterdam, The Netherlands. [2]Department of Physiology, Amsterdam UMC, Location Vrije Universiteit Amsterdam, Amsterdam, The Netherlands. [3]Amsterdam Cardiovascular Sciences, Amsterdam, The Netherlands. [4]Amsterdam Gastroenterology, Endocrinology and Metabolism, Amsterdam, The Netherlands. [5]United for Metabolic Diseases, Amsterdam, The Netherlands. [6]Emma Center for Personalized Medicine, Amsterdam UMC, Amsterdam, The Netherlands. [7]Department of Endocrinology and Metabolism, Amsterdam UMC Location University of Amsterdam, Amsterdam, The Netherlands. [8]Department of Pediatrics, Emma Children's Hospital, Amsterdam Gastroenterology, Endocrinology and Metabolism, Amsterdam UMC, Location University of Amsterdam, Amsterdam, The Netherlands. [9]Department of Human Genetics, Amsterdam Reproduction and Development, Amsterdam UMC Location University of Amsterdam, Amsterdam, The Netherlands. ✉E-mail: r.h.houtkooper@amsterdamumc.nl; s.m.nielsen@amsterdamumc.nl

**Glossary**

**Barth Syndrome (BTHS)**
is a rare, X-linked genetic metabolic disorder caused by mutations in *TAZ* encoding the enzyme tafazzin. Tafazzin plays a critical role in the remodelling of cardiolipin, a unique phospholipid essential for mitochondrial membrane integrity and function.

**Dilated cardiomyopathy (DCM)**
is a type of heart muscle disease where the heart chambers, especially the left ventricle, become dilated and the heart muscle weakens.

**Hypertrophic cardiomyopathy (HCM)**
is a genetic heart disease characterised by abnormal thickening (hypertrophy) of the heart muscle.

**IEMbase**
https://www.iembase.com/ is an online database representing a large class of rare inborn errors of metabolism (IEM). IEMbase serves as knowledge-base and a smart system (artificial intelligence) for curation and diagnosis support.

**Inherited metabolic diseases (IMD)**
complex class of conditions that can affect the metabolism of carbohydrates, amino acids, lipids, steroids but also nucleic acids, mitochondria and neurotransmitters.

**(h)iPSC**
human induced pluripotent stem cell are a type of pluripotent stem cell that can be generated directly from a somatic cell.

**Long-chain fatty acid oxidation disorders (lcFAODs)**
are a group of rare, inherited metabolic disorders that impair the body's ability to break down long-chain fatty acids.

**Metabolic cardiomyopathies**
heart muscle diseases resulting from metabolic disturbances, typically due to genetic defects in enzymes or proteins involved in cellular metabolism.

**OMIM**
https://omim.org/ is an online catalogue of human genes and genetic disorders.

and cell death (Kornfeld et al, 2015; Devalla and Passier, 2018). The human heart is rich in mitochondria as it requires 6 kg of ATP generated daily to ensure proper contractile function and has a very limited capacity for substrate storage (Pavez-Giani and Cyganek, 2021; Kolwicz et al, 2013). The heart's metabolic flexibility is essential to adapt to high energy requirements for constant contraction, and it relies mainly on fatty acid oxidation (FAO) for energy production (60–90%) as a stable and high energy substrate (Stanley et al, 2005). IMDs can lead to metabolic cardiomyopathies through three possible mechanisms: (1) the accumulation of toxic intermediates, (2) excessive substrate storage and infiltration, and (3) cardiac energy deficiency which can trigger maladaptive responses and structural remodelling, causing cardiac dysfunction (Fig. 1) (Cox, 2007).

Studies on metabolic cardiomyopathies aim to define the complex mechanisms that underlie defects in myocardial energy metabolism and how they cause and aggravate structural remodelling and dysfunction of the heart. As such, unravelling the metabolic alterations in metabolic cardiomyopathies can offer new perspectives for the more common cardiomyopathies linked to sarcomere variants, where in-depth analysis of the interplay between mitochondrial dysfunction and cardiac structural abnormalities is currently lacking. Here we discuss the clinical complexity and heterogeneity of metabolic cardiomyopathies with a focus on several IMDs with severe cardiac complications, namely long-chain fatty acid oxidation disorders (lcFAODs) and Barth Syndrome (OMIM 302060, BTHS). We focus on the current state-of-the-art experimental models, as well as the challenges and opportunities to study unknown pathophysiological mechanisms leading to diverse cardiac phenotypes in these metabolic cardiomyopathies. Lastly, we emphasise the use of 2D and 3D stem cell-based cardiomyocyte models that can be used to identify such mechanisms and the development of personalised human models and treatment options for these life-threatening diseases.

# Clinical heterogeneity in metabolic cardiomyopathies

Several groups of IMDs are associated with cardiomyopathies, including mitochondrial diseases, fatty acid oxidation disorders

(FAODs), glycogen storage diseases, lysosomal storage diseases, mucopolysaccharidoses, organic acidaemias and more (Cuenca-Gómez et al, 2024). The most common types of cardiomyopathies linked to IMDs are HCM and DCM but these can also include endocardial fibroelastosis, left-ventricular non-compaction, and restrictive cardiomyopathy and cardiac valve disease (Sadiku et al, 2025). Given this clinical heterogeneity, we provide two tables listing IMDs associated with cardiac symptoms. Table EV1 is a list of all IMDs for which at least one case presenting with cardiomyopathy has been reported, identified by searching IEMbase (Ferreira et al, 2019). Table 1 is a more detailed summary of myopathy-associated IMDs with commonly reported and well-described cardiac phenotypes. For these, we give the prevalence, mutated gene and summarise clinical phenotype—both cardiac and non-cardiac features.

Mitochondrial diseases are the most common IMDs with a prevalence of 1:5000 and are caused by pathogenic variants in either nuclear genes or in mitochondrial DNA (mtDNA) coding for proteins involved in mitochondrial respiratory chain functioning (Duran et al, 2019; Sulaiman et al, 2020). mtDNA variants can be present in different heteroplasmy levels in different issues, contributing to the wide variety of affected organs, and this clinical heterogeneity contributes to delayed diagnosis in many cases (Sulaiman et al, 2020). Among primary mitochondrial disorders such as mitochondrial encephalomyopathy, lactic acidosis, and stroke-like episodes (MELAS) and myoclonic epilepsy with ragged red fibers (MERRF) syndromes, the cardiac clinical phenotype includes HCM, DCM and cardiac conduction abnormalities (Wolff-Parkinson-White syndrome) (Table 1). Treatment options for these patients are limited and focused on symptom management and supportive measures.

FAODs have a wide range of clinical symptoms affecting several organs once fatty acids can no longer be converted into tricarboxylic acid cycle (TCA) cycle intermediates such as acetyl-CoA (Merritt et al, 2020). There is a wide range of clinical severity in FAODs, with limited genotype-phenotype correlation, however, some biochemical and functional measures (e.g. fatty acid oxidation flux) do correlate with disease severity in some of the disorders (Table 1). This is the case for carnitine palmitoyltransferase 2 deficiency (CPT2D), multiple acyl-CoA dehydrogenation deficiency (MADD) and very long-chain acyl-CoA dehydrogenase deficiency

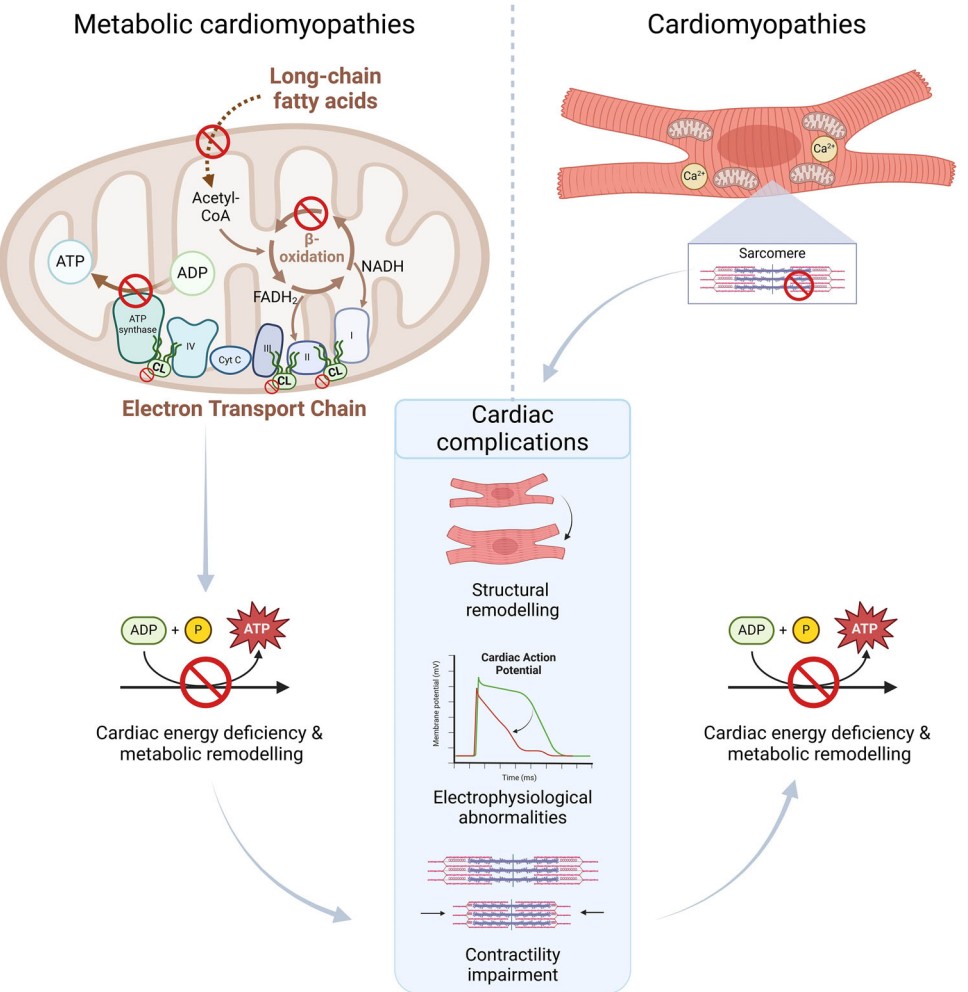

**Figure 1. Schematic view of the distinction between metabolic cardiomyopathies and cardiomyopathies such as HCM and DCM.**

In metabolic cardiomyopathies, as here illustrated with lcFAOD, mitochondrial dysfunction and metabolic remodelling are the primary cues for cardiac remodelling, resulting in complications such as structural disarray, contractility defects and electrophysiological abnormalities. In case of cardiomyopathies such as HCM and DCM, variants in sarcomere and calcium-handling proteins cause cardiomyocyte dysfunction which subsequently leads to secondary changes in the heart, including mitochondrial perturbations and metabolic remodelling. Graphics were created with BioRender.com.

(VLCADD) but not so clear for medium-chain acyl-CoA dehydrogenase deficiency (MCADD) and carnitine transporter deficiency (CTD) (Diekman et al, 2015; Olpin, 2013). FAODs prevalence estimation varies according to the subtype and there are currently no targeted treatment options (Knottnerus et al, 2018). In the more severe cases, cardiovascular treatment includes inotropic therapy, mechanical cardiac support and heart transplant (Merritt et al, 2020).

Glycogen Storage Diseases (GSDs) result from enzyme deficiencies impairing either the breakdown or the storage of glycogen, which leads to low blood glucose levels during fasting periods as well as organomegaly and dysfunction due to storage (Conte et al, 2023). The cardiac consequences can arise from glycogen breakdown deficiency, but it is mostly the accumulation of glycogen deposits that leads to cardiomyocyte vacuolation (Berling et al, 2021; Austin et al, 2012). The GSDs are subdivided into categories according to both genotypic and phenotypic heterogeneity. For example, acid maltase deficiency is classified in three forms:

"classic" Pompe disease (the infantile form), "non-classical" infantile Pompe disease (the childhood/juvenile form) and the adult form. Pompe disease occurs due to intralysosomal accumulation of glycogen, and the infantile form manifests with a very severe HCM phenotype with an estimated prevalence of 1:30,000–140,000 births worldwide (depending on the form) and an average life expectancy of 1 year (Conte et al, 2023). The "non-classical" infantile Pompe disease has a milder cardiac phenotype and the adult form presents without cardiac involvement in the majority of cases (Di Rocco et al, 2007). For Pompe disease patients, enzyme replacement therapy is a standard treatment that reverses the HCM phenotype (Lim et al, 2014; Kishnani et al, 2007).

Enzyme replacement therapy is also the established treatment for Fabry disease and Gaucher disease, which are lysosomal storage diseases (Cox, 2007; Conte et al, 2023). Both Fabry disease and Gaucher disease occur due to deficient lysosomal enzymes, α-galactosidase A and glucocerebrosidase, respectively, which are needed for proper degradation of complex lipids in the lysosomes

Table 1. Inherited metabolic disorders associated with cardiomyopathy.

| Disorder (OMIM#) | Prevalence | Gene (OMIM#) | Clinical phenotype | |
|---|---|---|---|---|
| | | | Cardiac | Non-cardiac |
| **Fatty acid metabolism disorders** | | | | |
| CACTD (212138) | 1:750,000–1:2,000,000 | SLC25A20 (613698) | Neonatal cardiomyopathy, ventricular arrhythmias | Depending on severity, can include hypoglycemia, hyperammonemia, myopathy |
| CPT2D (255110) | 1:600,000–1:2,000,000 | CPT2 (600650) | HCM (only in the severe childhood onset forms) | Depending on severity, can include exercise intolerance, rhabdomyolysis Severe forms: congenital anomalies (brain and kidney malformations) |
| CTD (212140) | 1:77,000 | SLC22A5 (603377) | Only in the severe forms: DCM, ventricular arrhythmias | Only in the severe forms: hypoketotic hypoglycemia, hyperammonemia, liver dysfunction, hypotonia |
| LCHADD (609016) | 1:250,000–1:750,000 | HADHA (600890) | HCM, DCM (cardiac function can fluctuate), arrhythmias, sudden death | Retinopathy, myopathy, peripheral neuropathy |
| MADD (231680) | 1:750,000–1:2,000,000 | ETFA (608053), ETFB (130410), ETFDH (231675) | Severe forms: Arrhythmias, HCM, sudden death | Milder forms: hepatomegaly, myopathy, rhabdomyolysis |
| MCADD (607008) | 1:15,000 | ACADM (201450) | Arrhythmias | Lethargy, hypoketotic hypoglycemia, liver dysfunction |
| MTPD (609015) | 1:250,000–1:750,000 | HADHA (600890), HADHB (143450) | HCM, DCM (cardiac function can fluctuate), arrhythmias, sudden death | Myopathy, retinopathy, peripheral neuropathy |
| VLCADD (201475) | 1:85,000 | ACADVL (609575) | HCM (severe forms only), DCM (cardiac function can fluctuate), arrhythmias, sudden death | Myopathy, hypoglycemia, encephalopathy |
| **Mitochondrial disorders** | | | | |
| BTHS (302060) | 1:1,000,000[a] | TAFAZZIN (300394) | Arrhythmias, DCM, HCM, left-ventricular non-compaction, endocardial fibroelastosis | Skeletal myopathy, neutropenia, growth delay, motor delay, fatigue, exercise intolerance, hypoglycemia, lactic acidosis, hyperammonemia, 3-methylglutaconic aciduria |
| Complex I Deficiency (252010) | Unknown | NDUFS4 (602694) | HCM, DCM | Encephalopathy, recurrent seizures, intellectual disability, ataxia, dystonia, developmental regression, lactic acidosis, hypotonia, hepatomegaly, kidney dysfunction, nystagmus, optic atrophy, apnoea, respiratory failure and insufficiency |
| Complex II Deficiency (252011) | 2–23% | SDHA (600857) | DCM with non-compaction, HCM | Psychomotor delay, seizures, lactic acidosis, liver disease, myopathy |
| Complex III Deficiency (124000) | Unknown | BCS1L (603647) | HCM | Lactic acidosis, hypotonia, hypoglycemia, failure to thrive, encephalopathy, and delayed psychomotor development |
| Complex IV Deficiency (220110) | Unknown | Multifactorial with both nuclear and mitochondrial variants | HCM | Delayed growth, hypotonia, encephalopathy, liver disease/failure |
| Complex V Deficiency (604273) | Unknown | ATPAF2 (608918) | HCM | Hypotonia, lactic acidosis, hyperammonemia |
| Kearns–Sayre syndrome (530000) | 1:125,000 | Mitochondrial DNA deletions/duplications | Cardiomyopathy, cardiac conduction defects | Chronic progressive external ophthalmoplegia, pigmentary retinopathy, ataxia |
| MELAS (540000) | 1:4000 | Mitochondrial transfer RNA variants (most common: MTTL1 (590050)) | HCM, Wolff-Parkinson-White syndrome, conduction defects, atrial fibrillation | Sensorineural deafness, myopathy, diabetes mellitus, stroke-like episodes |
| MERRF (545000) | Unknown | Mitochondrial DNA variants (most common: MT-TK (590060)) | HCM, DCM | Recurrent seizures, ataxia, myopathy, cognitive decline, optic nerve atrophy |
| Sengers syndrome (212350) | Unknown | AGK (610345) | HCM | Congenital cataracts, skeletal myopathy, exercise intolerance, lactic acidosis |

**Table 1.** (continued)

| Disorder (OMIM#) | Prevalence | Gene (OMIM#) | Clinical phenotype | |
|---|---|---|---|---|
| | | | Cardiac | Non-cardiac |
| **Glycogen metabolism disorders** | | | | |
| GSD IIa – Pompe disease (232300) | 1:18,711 | GAA (606800) | Severe HCM (infantile onset form) | Proximal myopathy, respiratory failure (late-onset forms) |
| GSD III – Cori disease (232400) | 1:100,000 | AGL (610860) | IIIa: Left ventricular hypertrophy, atrial and ventricular fibrillation, sudden death | Myopathy (IIIa), liver involvement (IIIa and IIIb), diabetes mellitus type II |
| GSD IV – Andersen disease (232500) | 1:600,000–1:800,000 | GBE1 (607839) | DCM | Neonatal death, hypotonia, skeletal myopathy, neurodegenerative disease, liver disease |
| PRKAG2 (261740) | Unknown | PRKAG2 (602743) | HCM (Wolff-Parkinson-White Syndrome), DCM, supraventricular tachyarrhythmias, sudden cardiac death | Skeletal myopathy |
| GSD IIb – Danon disease (300257) | Unknown | LAMP2 (309060) | HCM, severe conduction abnormalities | Skeletal myopathy, neurocognitive developmental delay |
| **Lysosomal storage disorders**[b] | | | | |
| Fabry disease (301500) | 1:40–117,000 | GLA (300644) | HCM, conduction disorders, arrhythmias, (diastolic) heart failure | Renal insufficiency, stroke, heat intolerance, neuropathic pain, airway obstruction |
| GM1 (230500) | 1:100–200,000 | GLB1 (611458) | HCM, DCM, congestive cardiomyopathy (infantile, juvenile and adult) | Developmental regression, exaggerated startle reaction, hepatosplenomegaly, seizures, intellectual disability, vision impairment, macular cherry-red spots, coarse facial features, enlarged gums, dysostosis multiplex, stiff joints, poor muscle tone |
| GM2 | 1:500,000–1:1,500,000 | GM2A (613109) | Cardiomyopathy, thickening of valves | Generalised neurodegeneration and mild skeletal changes |
| MPS I – Hurler-Scheie syndrome (607015) | 1:100,000 | IDUA (252800) | Cardiac valve disease, HCM, patent foramen ovale, endocardial fibroelastosis | Developmental delay, chronic hearing loss |
| MPS II – Hunter syndrome (309900) | 1:156,000 | IDS (300823) | Right and left ventricular hypertrophy, arrhythmias | Coarse facial features, joint stiffness and deformities, hearing loss, recurrent upper respiratory infections, enlarged tonsils and adenoids, developmental and intellectual delay, enlarged liver and spleen, hydrocephalus |
| MPS VI – Maroteaux-Lamy syndrome (253200) | 1:43,000– 1:1,500,000 | ARSB (611542) | Left-ventricular hypertrophy, conduction abnormalities | Short stature, hepatosplenomegaly, dysostosis multiplex, stiff joints, corneal clouding, facial dysmorphism |
| MPS VII – Sly syndrome (253220) | 1:300,000–1:2,000,000 | GUSB (611499) | Severe: valvular heart disease, cardiomyopathy | Developmental delay, macrocephaly, coarse hair and facial features, macroglossia, corneal clouding, hearing loss, neurocognitive impairment, frequent upper respiratory tract infections, skeletal abnormalities, recurrent ear infections, gingival hypertrophy, hepatosplenomegaly, umbilical hernias |
| **Peroxisomal disorder** | | | | |
| Refsum disease (266500) | 1:10,000,000 | PHYH (602026) | Arrhythmias, conduction defects, HCM, DCM | Retinitis pigmentosa, anosmia, peripheral neuropathy, cerebellar ataxia, sensorineural hearing loss, ichthyosis, and multiple epiphyseal dysplasia |

*BTHS* Barth syndrome, *CACTD* carnitine-acylcarnitine translocase deficiency, *CPT1D* carnitine palmitoyltransferase-1 deficiency, *CPT2D* carnitine palmitoyltransferase type 2 deficiency, *CTD* carnitine transporter deficiency, *DCM* dilated cardiomyopathy, *GM* Gangliosidosis, *GSD* glycogen storage disease, *HCM* hypertrophic cardiomyopathy, *LCHADD* long-chain 3-hydroxy acyl-CoA dehydrogenase deficiency, *MADD* multiple acyl-CoA dehydrogenase deficiency, *MCD* muscle carnitine deficiency, *MERRF* Myoclonic Epilepsy with Ragged-Red Fibres, *MPS* mucopolysaccharide storage disorders, *MTPD* mitochondrial trifunctional protein deficiency, *MELAS* mitochondrial encephalomyopathy, lactic acidosis, and stroke-like episodes, *RCM* restrictive cardiomyopathy, *SCADD* short-chain 3-hydroxyacyl-CoA dehydrogenase deficiency, *SPCD* systemic primary carnitine deficiency, *VLCADD* very long-chain acyl-CoA dehydrogenase deficiency.
[a]Prevalence reported in males (Taylor et al, 2022).
[b]Including GM1 and GM2 gangliosidoses.

(Chi et al, 2020). The accumulation of non-degraded substrates is generally responsible for the cardiac defects associated with lysosomal storage disorders. The accumulation of globotriaoslyceramide (Gb3) and its derivative lysoGb3 in Fabry disease leads to progressive cardiac hypertrophy, fibrosis, arrhythmias and ultimately heart failure (Linhart et al, 2020). Gaucher disease with a global prevalence of 1:60,000 births (Cuenca-Gómez et al, 2024) more rarely affects the heart but the accumulation of glucocerebrosides in immune cells can infiltrate the heart, and there have been reports on calcification of the heart valves (Abrahamov et al, 1995) and a rare restrictive cardiomyopathy case associated with Gaucher disease (Kundu et al, 2021).

This review focuses on lcFAODs and BTHS, diseases that primarily relate to lipid metabolism and for which our institute has a long tradition and serves as an expertise centre. We describe clinical manifestations and current treatment options for lcFAODs and BTHS. In addition, we refer to cardiac disease mechanisms that have been investigated using different disease models, discuss the models' shortcomings and the valuable insights they have provided into the heterogeneity of cardiac pathophysiology.

# Long-chain fatty acid oxidation deficiencies

LcFAODs are a group of rare, autosomal recessively inherited IMDs. They are systemic disorders with significant morbidity and mortality caused by genetic defects in enzymes involved in mitochondrial FAO40. lcFAODs refer to deficiencies impairing either the transport of long-chain fatty acids into the mitochondria or the fatty acid breakdown in the mitochondrial matrix. Collectively, lcFAODs occur with an incidence of approximately 1 in 9300 individuals in the United States, Australia and Germany, although incidence varies depending on ethnic background of the population (Lindner et al, 2010).

### Genetics and clinical manifestations

Long-chain fatty acids contain more than 12 carbons and are transported into the mitochondria through the carnitine shuttle40. The first reaction in this transport is catalysed by carnitine palmitoyltransferase 1 (CPT1), which is followed by the import of acylcarnitines across the inner mitochondrial membrane by carnitine acylcarnitine translocase (CACT). Carnitine palmitoyltransferase 2 (CPT2) is the last component of the carnitine shuttle, releasing free acyl-CoAs into the mitochondrial matrix to enter the β-oxidation spiral (Merritt et al, 2020; Knottnerus et al, 2018; El-Gharbawy and Vockley, 2018). The β-oxidation spiral consists of four sequential enzymatic reactions. The first step involves different enzymes depending on chain length, such as very long-chain acyl-CoA dehydrogenase (VLCAD), medium-chain acyl-CoA dehydrogenase (MCAD) and short-chain acyl-CoA dehydrogenase (SCAD). The other three reactions in the β-oxidation spiral are mediated by the mitochondrial trifunctional protein (MTP), which contains the long-chain enoyl-CoA hydratase (LCEH), long-chain-3-ketoacyl-CoA thiolase (LCKAT) and long-chain 3-hydroxyacyl-CoA dehydrogenase (LCHAD) domains to complete the oxidation cycle of long-chain fatty acids, and release a molecule of acetyl-CoA for further oxidation in the TCA cycle (Knottnerus et al, 2018). This review covers CPT2 deficiency (CPT2D), CACT deficiency (CACTD), VLCAD deficiency (VLCADD), MTP deficiency (MTPD), and LCHAD deficiency (LCHADD).

A wide spectrum of clinical symptoms is associated with lcFAODs, and phenotypic heterogeneity is also present among patients carrying the same pathogenic variants. Cardiac manifestations first occur most often in the postnatal period and less often later on in life, and include HCM or DCM as well as cardiac arrhythmias. Cardiac disease can be triggered by energy deficiency and/or the accumulation of lcFAO intermediates in the cytoplasm, especially in circumstances in which reliance on fatty acids as the main myocardial energy substrate is increased.

Non-esterified fatty acids can activate voltage-dependent calcium currents in cardiomyocytes. The oscillatory and non-oscillatory potentials result in cytotoxic calcium overload and, ultimately, arrhythmias (Bonnet et al, 1999). The intermediates accumulated in the cytoplasm and lysosomes of cardiomyocytes trigger inflammation and contribute to an increase in tissue size. Consequently, cardiac myofibrils are misaligned and, ultimately, contribute to cardiac remodelling (Cox, 2007; McCoin et al, 2019). Energy insufficiency can affect ATP hydrolysis by the myosin heads. This is required to generate the necessary force to slide the thick filaments past the actin-thin filaments during contraction (Cox, 2007; Merritt et al, 2020).

Symptom onset in lcFAODs' can be unpredictable and sudden, and significant mortality is still observed, especially in the first years of life (Pannier, 2020). Symptom onset can be triggered by reduced energy intake, increased physical activity, and infections (Merritt et al, 2020; El-Gharbawy and Vockley, 2018). Symptoms include hypoglycemia, liver dysfunction, rhabdomyolysis, skeletal myopathy, and cardiomyopathy. These symptoms can lead to hospital admissions and require specialised care (Merritt et al, 2020). For the milder forms of, for example, VLCADD or CPT2 deficiency, symptom onset is often in late childhood or adolescence and can be more gradual (exercise intolerance/myalgia) or with a first episode or rhabdomyolysis without other organ involvement. A genotype-phenotype association has been described in some of these diseases, with missense variants being associated with milder phenotypes compared to null variants (Spiekerkoetter, 2010). In VLCADD patients, ACADVL missense variants were associated with mild phenotypes (Goetzman et al, 2007). For CACTD, MTPD and the infantile form of CPT2D, there are no clear genotype-phenotype correlations reported (Spiekerkoetter, 2010). We and others have developed advanced biochemical assays to distinguish mild and severe cases of lcFAODs with the aim of predicting the clinical outcome at an early stage (Gregersen et al, 2001; Andresen et al, 1999; Knottnerus et al, 2020b; Diekman et al, 2015).

For early patient identification and management, lcFAODs were included in newborn screening (NBS) programs in many countries (Merritt et al, 2020; Pannier, 2020) improving some clinical outcomes (e.g. hypoglycaemia in VLCADD), but certainly not all (limited effect on muscle symptoms and cardiomyopathy cannot always be prevented) (Bleeker et al, 2019).

### Clinical management

Management of lcFAOD patients is mainly focused on timely diagnosis followed by dietary and lifestyle recommendations (Merritt et al, 2020; Vockley, 2020). Specifically for lcFAODs, nutritional management is centred around reducing fasting time,

providing alternative energy substrates and restricting the intake of long-chain fats, thus limiting the supply to the (partially) defective β-oxidation pathway (Vockley, 2020; Merritt et al, 2018). Indeed, where possible, the predicted disease severity dictates to which extent dietary fat intake should be restricted (Merritt et al, 2018).

It is worth noting that cardiomyopathy can be reversed in many cases (Cox, 2007; Spiekerkoetter, 2010). Positive results have been described for a combination of optimising energy intake and supplementing medium-chain fatty acids, either even- or odd-chain (C7). Triheptanoin is a triglyceride oil that contains heptanoate (C7:0) side chains and is approved by the FDA as a drug for the treatment of lcFAODs (Dojolvi®). Following lipolytic release of the heptanoate, this medium-chain fatty acid can undergo two cycles of β-oxidation releasing acetyl-CoA, and a molecule of propionyl-CoA which can feed the TCA cycle. Heptanoate can be processed into 5-carbon ketone bodies (Merritt et al, 2018), and these can be used as an alternative fuel source for the skeletal muscle and the myocardium, bypassing the deficient enzymes (Merritt et al, 2020). Analysis of the effectiveness of triheptanoin has been hampered by disease heterogeneity and differences in disease manifestations throughout life (e.g., the number of hypoglycaemic events generally goes down after the first year of life). Other complicating factors include: lack of knowledge on natural disease course, lack of standardised clinical endpoints, effect of other factors such as overall diet and exercise. In addition, triheptanoin cannot be used for all lcFAODs (Norris et al, 2021). For instance, for MADD patients, medium-chain triglycerides (MCTs), including triheptanoin, are not advised given that the β-oxidation of all fatty acids, not only long-chain fatty acids, is compromised. As an alternative, ketone bodies such as β-hydroxybutyrate and acetoacetate could work as an oxidative substrate for cardiac and skeletal muscle and can be metabolised by all lcFAOD patients, including MADD patients. Studies reported improved cardiac function after ketone salts administration to MADD patients (Van Hove et al, 2003). Ketone ester supplementation has been studied in VLCADD patients. After administration of a single dose of ketone ester, long-chain acylcarnitine accumulation in the cytoplasm was significantly lower during exercise compared to isocaloric carbohydrate administration, though possible cardiac effects have not been studied yet (Bleeker et al, 2020).

Other treatments such as resveratrol (a dietary polyphenol) and bezafibrate (a lipid-lowering agent) have been studied given their potential to enhance FAO flux by inducing FAO and other mitochondrial enzymes. This is especially relevant in milder disease phenotypes with residual FAO enzyme activity (Yamaguchi et al, 2012; Aires et al, 2018; Bastin et al, 2011; Djouadi et al, 2003). Pre-clinical studies reported a beneficial effect of resveratrol on mRNA level in CPT2D fibroblasts (Aires et al, 2018). In addition, exposure to resveratrol normalised FAO in cells from CPT2D and VLCADD patients with a mild phenotype (Bastin et al, 2011). Bezafibrate lowered the elevated long and medium-chain acylcarnitines accumulated in VLCADD, CPT2D, CACTD, and MTPD fibroblasts (Yamaguchi et al, 2012) and corrected FAO in mild CPT2D fibroblasts (Djouadi et al, 2003). However, in a clinical study, daily resveratrol intake did not change palmitate oxidation, heart rate or exercise capacity of VLCADD and CPT2D patients after exercise (Storgaard et al, 2022). An open-label, non-randomised, multi-centre study on five VLCADD patients suggested improvement of quality of life and reduction of hospitalisations (Shiraishi et al,

2021). Nonetheless, in a randomised, double-blind, crossover clinical trial (2008-006704-27), bezafibrate was administered for 3 months in 5 CPT2D and 5 VLCADD patients, and this did not affect palmitate oxidation and FAO at rest or during exercise (Vissing et al, 2013). Thus, although mitochondrial biogenesis inducers have shown great potential as a treatment option for FAODs in pre-clinical models, so far no clinically relevant effect has been established.

## Experimental models

Despite challenges in developing clinically significant cardiac research models due to the multifactorial nature of lcFAODs, successful studies have provided valuable insights into disease pathophysiology and therapeutic opportunities. For VLCADD, mice models have been generated that are knockout (KO) for *Acadvl* (Exil et al, 2003; Gélinas et al, 2011; Cox et al, 2001; Bakermans et al, 2013; Xiong et al, 2014; Gaston et al, 2020; Tucci et al, 2012; Cox et al, 2009; Tucci et al, 2014; Veld et al, 2009; Liebig et al, 2006; Spiekerkoetter et al, 2005). Mice, unlike humans, also express *Acadl* coding for the long-chain acyl-CoA dehydrogenase (LCAD) enzyme. LCAD has overlapping substrate preference with VLCAD and can partially compensate for its loss (Chegary et al, 2009). Therefore, other studies have used *Acadl* knock-out (KO) mice79–82, and even mice in which *Acadl*-KO was combined with a heterozygous state of *Acadvl* that partly mimicked VLCADD in humans (Diekman et al, 2014). The phenotype of *Acadl*-KO mice resembled the human VLCADD phenotype (Kurtz et al, 1998). This was corroborated later on when the study of the phenotypic differences between mice and humans highlighted how these *Acadl*-KO mice more closely depict the human VLCADD phenotype than *Acadvl*-KO mice (Chegary et al, 2009). Pathogenic variants in *Acadvl* were linked to mild transient myocardial hypertrophy (Cox et al, 2009), and DCM was also identified in VLCADD mice with an increase in left ventricular (LV) end-diastolic and end-systolic dimensions and irregular cardiac histology (Xiong et al, 2014). Electrophysiology abnormalities were detected in *Acadvl*-KO mice, such as induced ventricular arrhythmias (Exil et al, 2003), prolonged corrected QT (QTc) interval (Gélinas et al, 2011), and severe bradycardia upon physiological stress (Exil et al, 2006). Furthermore, chronic energy deficiency led to reduced ejection fraction in 1-year-old VLCADD mice, which progressed over time and showed features similar to diabetic heart disease (Tucci et al, 2014). Another study on VLCADD mice showed distinct compensatory mechanisms in specific tissues. Particularly, in the heart and liver, but not skeletal muscle, they observed the upregulation of acyl-CoA dehydrogenase (ACAD) for long-chain acyl-CoA substrates oxidation (Tucci et al, 2012).

In addition to VLCADD, mouse models have also been developed for other lcFAODs. Cardiac hypertrophy triggered by CPT2D was examined in *Cpt2* KO mice. In this study, dietary octanoate was applied, and tissue-specific compensatory mechanisms were identified. Cardiac hypertrophy was not rescued and, consequently, cardiac function showed no improvement (Pereyra et al, 2021). Hadha-KO mice to model MTPD demonstrated severe cardiomyocyte degeneration and neonatal death (Ibdah et al, 2001). No follow-up intervention studies were performed with this KO model. The first viable MTPD model were mice with the p.M404K missense variant that significantly decreased the HADHA and

HADHB protein levels, and these mice survived into adulthood. In the M404K mice, cardiac arrhythmias were detected at 6 months of age along with focal fibrosis in the entire heart and sarcomere loss (Kao et al, 2006). A knock-in (KI) mouse model of the human variant c.1528 G > C in Hadha was generated using CRISPR/Cas9, and it successfully resembled the LCHADD human phenotype, showing eccentric hypertrophy indicative of a DCM phenotype (Gaston et al, 2023). In conclusion, these mouse models have provided valuable insights into mild and severe cardiovascular manifestations, and we summarise the specific cardiac symptoms that were studied in each model in Table 2.

Despite the important progress that has been made using simple model organisms and mice, major physiological differences to humans still impose challenges when translating the model findings to patients. lcFAOD mouse models often have high mortality rates leading to major complications to study long-term pathology and disease mechanisms. For instance, MTPD and CPT1D mice models have a high neonatal and embryonic mortality, respectively (Ibdah et al, 2001; Nyman et al, 2005; Ji et al, 2008). Moreover, a zebrafish model of MADD with *etfa* inactivated was used to investigate disease progression, and 20% of mutants showed cardiac oedema and died 5–6 days after fertilisation (Kim et al, 2013).

# Barth syndrome

BTHS was first described by Prof. Peter Barth in 1983 (Barth et al, 1983). It is a rare X-linked inherited mitochondrial disorder affecting males, caused by pathogenic variants in TAFAZZIN (prevalence 1:1,000,000) (Taylor et al, 2022). The disease onset is in the infantile period, and the clinical spectrum is broad, including cardiomyopathy (73% of cases), skeletal myopathy, neutropenia, and growth and developmental delay (Taylor et al, 2022). The rate of disease progression is highly variable between patients, and there are currently around 250 known cases worldwide, with cardiac disease being the main driver of outcomes92. Infancy and early childhood are particularly high-risk periods for cardiac death. Bacterial infections due to neutropenia are the second driver of mortality in BTHS (Taylor et al, 2022).

## Genetics and clinical manifestations

On a molecular level, BTHS is caused by pathogenic variants in *TAFAZZIN* on chromosome Xq28 (Taylor et al, 2022). This gene encodes the tafazzin protein, which is a transacylase enzyme that transfers fatty acyl groups from phospholipids to monolysocardiolipin (MLCL) to generate mature cardiolipin (CL). CL is a unique phospholipid that is specifically located in the mitochondrial membrane. Following primary synthesis, the immature CL undergoes further remodelling, which involves the transient formation of the intermediate MLCL (Houtkooper and Vaz, 2008). In case of BTHS, with Tafazzin deficiency, CL remodelling and maturation are impaired, leading to CL deficiency and accumulation of MLCL, which makes the increased MLCL/CL ratio the diagnostic marker used for BTHS (Vaz et al, 2022).

CL is mostly present at the inner mitochondrial membrane. This membrane forms the cristae structures: invaginations which incorporate the protein complexes of the electron transport chain (ETC). Consequently, CL is crucial for the structural stabilisation of the inner mitochondrial membrane and for the integrity and enzymatic activity of the ETC supercomplexes, facilitating OXPHOS towards ATP production (Houtkooper and Vaz, 2008). BTHS-related depletion of mature CL content leads to mitochondrial abnormalities, including unstable OXPHOS complexes, which can have detrimental effects on overall cardiac performance through increased oxidative stress, sarcomere disorganisation and reduced contractility (Wang et al, 2014). In addition, increased reactive oxygen species (ROS) production results in structural and functional damage of the mitochondria and ATP depletion (Zegallai and Hatch, 2021). In BTHS cardiac tissue analysed by electron microscopy, cardiomyocyte mitochondria show structural damage evident from increased size and abnormal cristae, and cardiomyocytes show intracellular vacuolates (Dudek and Maack, 2022).

Early diagnosis is crucial for optimal clinical monitoring and cardiac disease management, though phenotypic variability and absence of targeted treatment options represent major challenges for clinicians (Thompson et al, 2022). There is no genotype-phenotype correlation for BTHS, and variation in symptoms has been observed even within males of the same family (Ronvelia et al, 2012). Cardiac manifestations are highly variable and can range from endocardial fibroelastosis, a rare heart disorder that affects infants and children, to DCM, HCM, restrictive cardiomyopathy, left-ventricular non-compaction (LVNC) cardiomyopathy, and sudden cardiac death due to ventricular arrhythmias (Taylor et al, 2022). Importantly, cardiac disease is less likely to be the primary manifestation after 10 years of age, and ventricular arrhythmias can occur at times of apparent stable health (Clarke et al, 2013). Ventricular fibrillation and tachycardia contribute considerably to BTHS mortality and appear to be independent of the severity of cardiomyopathy (Thompson et al, 2022; Spencer et al, 2005). Even though cardiomyopathy can manifest in infancy, neutropenia and skeletal myopathy can also be identified first (Spencer et al, 2006). Neutropenia is a highly common clinical feature present in 70–86% of patients, and severe infections are the presenting manifestation in 18% of patients (Taylor et al, 2022).

## Clinical management

As no cure is available for BTHS, the current treatment strategy for BTHS patients is focused on specific organ symptom management. The strategy adopted for cardiac monitoring and treatment must be tailored to the individual, as the cardiac phenotype can evolve over time and can fluctuate in an unpredictable manner (Thompson et al, 2022). For example, a HCM phenotype can change into a DCM phenotype (Kang et al, 2016) or LVNC presents in conjunction or isolated from DCM (Reynolds, 2015). For arrhythmia management, long-term surveillance is advised for all patients. The implant of an implantable cardioverter defibrillator (ICD) can be recommended upon detection of potentially life-threatening ventricular arrhythmias. Heart failure is treated with standard medication such as beta-blockers, angiotensin-converting enzyme (ACE) inhibitors, angiotensin II receptor blockers (ARBs), SGLT2 inhibitors and diuretics. In two separate short-term observational studies in the USA (34 patients) and UK (27 patients), patients were evaluated and cardiac function was stable, with no significant changes in LV size and systolic function, over 2 and 3 years, respectively. In both the USA and the UK study, the

**Table 2. Cardiac disease characteristics studied in different metabolic cardiomyopathy models.**

| | VLCADD | | | | CPT2D | MTPD | | | LCHADD | BTHS | | | | | |
|---|---|---|---|---|---|---|---|---|---|---|---|---|---|---|---|
| **Model** | LCAD-KO | VLCAD-KO | Combined LCAD-KO VLCAD+/− | Patient-derived-hiPCM-CMs | CPT2-KO | MTPa-KO | M404K | hiPSC-CMs | c.1528 G>C | TAFAZZIN-KD | TAFAZZIN-KO | TAFAZZIN-KI | hiPSC-CMs KO | hiPSC-CMs KI | Patient-derived-hiPCM-CMs |
| **Origin** | Mouse | Mouse | Mouse | Human | Mouse | Mouse | Human | Human | Mouse | Mouse | Mouse | Mouse | Human | Human | Human |
| Electrophysiological abnormalities | X | | | X | | | X | X | X | X | X | | X | | |
| Toxic intermediates accumulation | X | X | X | X | | | X | X | X | | | | | | |
| Cardiac dilation | LV | | | | | | | | X | LV | LV | | | | |
| Cardiac hypertrophy | LV (severe) | X (mild) | X | | LV | | | | | LV | | | | | |
| Diastolic dysfunction | X | X | | | | | | | | X | | X | | | |
| Contractile dysfunction | | | | | | | | | | | | | X | | X |
| LVNC | | | | | | | | | | X | | | | | |
| Sarcomere structural defects | | | | | | X | X | X | | | | | | | X |
| Abnormal mitochondrial structure | X | X | | | | X | X | X | | X | X | | | | X |
| Mitochondrial dysfunction | X | | | | | | | | | X | X | | X | X | X |
| **References** | Cox et al, 2001 | Exil et al, 2003 | Diekman et al, 2014 | Knottnerus et al, 2020a | Pereyra et al, 2021 | Ibdah et al, 2001 | Kao et al, 2006 | Miklas et al, 2019 | Gaston et al, 2023 | Acehan et al, 2011 | Zhu et al, 2021 | Chowdhury et al, 2023 | Liu et al, 2021 | Chowdhury et al, 2023 | Dudek et al, 2013 |

| | MELAS | | Pompe Disease | | | Danon disease | | | | | | Fabry disease | | |
|---|---|---|---|---|---|---|---|---|---|---|---|---|---|---|
| **Model** | Mito-mice tRNA$^{Leu(UUR)}$ 2748 | Patient-derived-hiPCM-CMs | GAA-KO | GAA-KI (infantile-onset) | Patient-derived-hiPCM-CMs | LAMP-2$^{−/−}$ | LAMP2 exon 6 deletion | LAMP-2 KO | LAMP-2 KO | Patient-derived-hiPCM-CMs | LAMP2$^{y/−}$ | GlaKO | G3Stg/GlaKO | Patient-derived-hiPCM-CMs |
| **Origin** | Mouse | Human | Mouse | Mouse | Human | Mouse | Mouse | Mouse | hiPSC | Human | Rat | Mouse | Mouse | Human |
| Electrophysiological abnormalities | X | X | X | X | X | X | X | | X | | | | | X |
| Toxic intermediates accumulation | | | X | | | X | X | | | | X | X | X | X |
| Cardiac dilation | | | | | | | | | | | | | | |
| Cardiac hypertrophy | X | X | X | X | X | X | X | X | | | X | | X | X |
| Diastolic dysfunction | | | | | | | | | | | | X | | |
| Contractile dysfunction | X | | | | | | | | | | | X | | X |

**Table 2.** (continued)

| Model / Origin | MELAS | | Pompe Disease | | | Danon disease | | | | | | Fabry disease | | |
|---|---|---|---|---|---|---|---|---|---|---|---|---|---|---|
| | Mito-mice tRNA$^{Leu(UUR)}$ 2748 / Mouse | Patient-derived-hiPCM-CMs / Human | GAA-KO / Mouse | GAA-KI (infantile-onset) / Mouse | Patient-derived-hiPCM-CMs / Human | LAMP-2$^{-/-}$ / Mouse | LAMP2 exon 6 deletion | LAMP-2 KO | LAMP2$^{y/-}$ / Rat | Patient-derived-hiPCM-CMs / Human | LAMP-2 KO hiPSC | GlaKO / Mouse | G3Stg/ GlaKO | Patient-derived-hiPCM-CMs / Human |
| LVNC | | | | | | | | | | | | | | |
| Sarcomere structural defects | | | X | | X | X | | | X | | X | X | X | X |
| Abnormal mitochondrial structure | X | | | | X | | | X | | X | | | | |
| Mitochondrial dysfunction | X | X | | | X | | | X | | X | X | | | X |
| References | Tani et al, 2022 | Ryytty et al, 2022 | Bijvoet et al, 1999 | Kan et al, 2022 | Huang et al, 2011 | Tanaka et al, 2000 | Yadin et al, 2022 | Hashem et al, 2017 | Ma et al, 2018 | Del Favero et al, 2020 | Barndt et al, 2023 | Kugadas et al, 2024 | Kugadas et al, 2024 | Birket et al, 2019 |

majority of participants were treated for heart failure (Spencer et al, 2006; Kang et al, 2016). In contrast, in some cases, mechanical support and cardiac transplantation are needed because of progressive heart failure (Li et al, 2021).

Aiming to target the specific consequences of the BTHS genetic defect, CL deficiency and mitochondrial dysfunction, new therapies have been recently explored in pre-clinical models such as AAV9 TAFAZZIN gene therapy (Tafazzin knock-down (Tafazzin-KD) mouse model107), bromoenol lactone (BTHS hiPSC-CMs (Wang et al, 2014)), arginine plus cysteine supplementation (BTHS hiPSC-CMs (Wang et al, 2014)), linoleic acid (BTHS hiPSC-CMs (Wang et al, 2014)), bezafibrate (Tafazzin-KD mouse model (Huang et al, 2017)) and resveratrol (Tafazzin-KD mouse model (Cole et al, 2020)). Following pre-clinical research, an open-label extended phase 2/3 clinical trial was initiated in 2017 to evaluate the effect of SS-31 specifically in BTHS patients (TAZPOWER trial, NCT03098797) (Sabbah, 2021). SS-31 or elamipretide is a tetrapeptide capable of crossing the mitochondrial outer membrane and associating with CL. As such, membrane stability can be improved and ATP production restored along with a decrease in ROS production. In the double-blind randomised phase of a study on the effects of Elamipretide in 12 BTHS patients, no statistically significant effect on the primary endpoints was reported. In the 36-week extension (completed in 8 out 12 patients), improvements on the 6-min walk test (6MWT) and BTHS Symptom Assessment (BTHS-SA) scale were reported, but the report on 2/3 of the randomized cohort complicates this analysis (Thompson et al, 2021). More recently, the final endpoint of the 168-week open-label extension phase was published, again reporting on 8/10 patients who were originally included in the trial. A significant cumulative improvement on 6MWT was reported, one of the primary endpoints of the original study (Thompson et al, 2024). Biochemically, MLCL/CL ratios improved. At cardiac assessment, increases in LV end-systolic and end-diastolic volumes and LV stroke volume were observed, though values were within the reference range at baseline, thus the clinical relevance of these observations needs to be clarified in future studies. Since several patients will have gone through their growth spurt during the trial (age at start open-label extension: 19.0 (±7.2) years), this may have contributed to these cardiac changes. Uncertainty about the clinically significant benefits of Elamipratide is reflected in the lack of unanimity of the FDA's Cardiovascular and Renal Drugs Advisory Committee members' vote for drug approval. Following this vote, a new study was designed to include a natural history control group to facilitate the evaluation of the treatment effects given the low disease prevalence inherent to BTHS (Hornby et al, 2022).

CARDIOMAN is the clinical trial aiming to test the effect of bezafibrate on mitochondrial biogenesis and the MLCL/CL ratio in BTHS patients (ISRCTN58006579) (Dabner et al, 2021). Bezafibrate is a PPAR pan-agonist which has been shown to ameliorate cardiomyopathy in a mouse model of Barth syndrome. Specifically, it ameliorates the development of LV systolic dysfunction in stressed Tafazzin-KD mice (Huang et al, 2017). The CARDIOMAN trial included eleven patients who took bezafibrate and a placebo for 15 weeks in random order, with a 1-month washout between periods. Study results have not been published in a scientific peer-reviewed journal yet, but a summary of results in laymen's terms is available online. Two additional clinical trials have focused on the

improving the reduced muscle strength in BTHS patients through resistance exercise training (RET) (Bittel et al, 2018; Bohnert et al, 2021). RET was well-tolerated by the BTHS patients included in the first 12-week trial, but cardiac parameters including fractional shortening and diastolic function remain unchanged (Bittel et al, 2018). In a separate study, in which another group of five BTHS patients performed RET sessions over 12 weeks while consuming 42 g/day of whey protein, LV mass and systolic function did not present any significant changes (Bohnert et al, 2021).

## Experimental models

While many of the initial studies on BTHS were performed on BTHS patients' fibroblasts (Vreken et al, 2000; Houtkooper et al, 2009), we now have access to additional experimental models, including yeast, fruit fly, zebrafish, and the mouse. Some of the earlier work has focused on simple model organisms such as yeast (Gu et al, 2004), fruit flies (Xu et al, 2009) and zebrafish (Khuchua et al, 2006), which recapitulate some BTHS features. In 2006, a morpholino tafazzin knock-down (tafazzin-KD) zebrafish model presented reduced heart rate and contractility and abnormal cardiac looping. In addition, embryonic lethality was associated with the tafazzin-KD when compared to wild-type controls (Khuchua et al, 2006). More recently, Tafazzin mutants were generated in Drosophila, which partly recapitulate the BTHS phenotype, including the elevated MLCL/CL ratio, reduced endurance and decreased respiratory control ratio, an indicator of mitochondrial health133. Interestingly, nicotinamide riboside (NR), a precursor to nicotinamide adenine dinucleotide (NAD+), improved mitochondrial function in this model system. Supplementing mutant flies with NR for 5 days increased their endurance compared to untreated controls, along with improved respiratory control ratio and mitochondrial number (Damschroder et al, 2022). However, the evolutionary distance to humans is a major disadvantage when using these models, which made it necessary to transition to mammalian models to study pathogenic mechanisms that are not well conserved between other species and humans (Pu, 2022).

Mice are most commonly used to study the cardiac pathophysiology of BTHS (Pu, 2022). Three mouse models have been developed for BTHS, including Tafazzin-KD (Suzuki-Hatano et al, 2019; Acehan et al, 2011; Soustek et al, 2011; Phoon et al, 2012; Greenwell et al, 2021), Tafazzin knock-out (Tafazzin-KO) (Zhu et al, 2021; Liu et al, 2021; Tomczewski et al, 2023), and very recently, a mouse model that expresses a specific patient variant (G197V) through targeted knock-in (Tafazzin-KI) (Chowdhury et al, 2023) (Table 2). Tafazzin-KD murine models showed a systemic reduction of 80–90% of Tafazzin gene expression by doxycycline-inducible short hairpin RNA (shRNA)-mediated Tafazzin-KD (Acehan et al, 2011; Soustek et al, 2011). The Tafazzin-KD mice presented cardiac mitochondrial abnormalities at 8 months (Acehan et al, 2011) and developed mild cardiac dysfunction at a late-onset, with cardiomyopathy becoming evident at 7–10 months (Soustek et al, 2011). A different study using the same model with a higher dose of doxycycline induction identified myocardial thinning along with diastolic dysfunction in foetal Tafazzin-KD hearts. Foetal cardiomyocytes displayed abnormal mitochondrial structure and disoriented mitochondria and sarcomeres (Phoon et al, 2012). In addition, glucose oxidation rates were reduced in Tafazzin-KD mice hearts at 8–10 weeks of age (Greenwell et al, 2021). Tafazzin-KO mice were recently characterised at 3, 6 and 12 months old, with lower heart weights

observed at all 3 ages when compared to age-matched wild-type control mice (Tomczewski et al, 2023). They also show fibrosis and severe cardiac dysfunction at 4 months old (Wang et al, 2020). Adeno-associated virus serotype 9 (AAV9)-Tafazzin gene-replacement improved the cardiolipin profile in Tafazzin-KO mice, and rescued the cardiac phenotype at high cardiomyocyte transduction (Wang et al, 2020). A cardiomyocyte-specific Tafazzin-KO model (Tafazzin cKO) was developed and presented DCM at 4 months old with significantly dilated LV chambers. When analysing isolated mitochondria from Tafazzin cKO mice, ROS and superoxide levels were elevated when compared to control (Zhu et al, 2021). A different study reported on the higher frequency of arrhythmias present in this model (Liu et al, 2021). Recently, a patient-variant KI BTHS mouse model was generated, where CL deficiency was associated with respirasome defects resulting in decreased respiratory capacity and increased ROS levels. In addition, measurements at the RNA and protein level revealed that mice heart tissue samples underwent a metabolic switch from FAO to glycolytic metabolism (Chowdhury et al, 2023).

In Tafazzin-KD mouse models, doxycycline was administered along with standard chow in their diet as the doxycycline-inducible shRNA-mediated Tafazzin-KD was shown to induce an 85%–90% silencing of Tafazzin in the heart (Acehan et al, 2011). This can be seen as an experimental limitation since this compound has been shown to impair diastolic function along with mitochondrial complex 1, reducing OXPHOS capacity (Wüst et al, 2021), which can be a confounding factor when interpreting the cardiac measurements on these models. Tafazzin can have 3–10% residual mRNA with 45% levels of tafazzin protein compared to controls in Tafazzin-KD mouse model, while most patients show lower expression and/or a variant in the catalytic domain. Conversely, Tafazzin-KO mice have no expression at all, which is also not in line with the patient context and post-neonatal survival posed a significant challenge (Wang et al, 2020). The new Tafazzin-KI mouse model could overcome some of these limitations. Nonetheless, important physiological differences between mice and humans may hinder cardiomyopathy investigations and effective prediction of drug responses on the basis of these models (Cox et al, 2009). Interspecies discrepancies include cardiac electrophysiology (e.g., heart rates of 600 beats per minute (mice) versus 75 beats per minute (humans)), metabolic pathways (LCAD and VLCAD functional overlap in mice (Houten et al, 2013)) and sex-specific phenotype differences (Goetzman, 2011). New models are warranted to capture the heterogeneity in cardiac phenotype and disease severity observed among metabolic cardiomyopathy patients (Table 2).

# Human stem cell-derived models for a personalised approach to metabolic cardiomyopathies

Further research is required to investigate the pathological mechanisms in humans that lead to specific tissue phenotypes and to identify novel therapeutic opportunities for metabolic cardiomyopathies. Several therapy options under investigation (i.e., bezafibrate, MCT oil) are still controversial as they lack robust studies with larger patient cohorts (Sacchetto et al, 2019; Vockley,

2020; Norris et al, 2021; Spiekerkoetter et al, 2010). Animal models have been considered the most physiologically relevant, but as described above, pose limitations with unpredictable outcomes (van der Velden et al, 2022; Novakovic et al, 2014). The need to achieve a better recapitulation of human cardiac physiology led to the development of novel cardiac cell culture models to investigate disease-causing mechanisms, potential therapies, and drug repurposing for metabolic cardiomyopathies.

## Development of human induced pluripotent stem cell-derived cardiac models

Human induced pluripotent stem cell (hiPSC) technology has been a key tool to develop novel cardiac models that mimic relevant disease parameters that are present in patients (Wang et al, 2014; Knottnerus et al, 2020a; Sacchetto et al, 2020; Mosqueira et al, 2018; Ceholski et al, 2018; Cohn et al, 2019). hiPSCs and hiPSC-derived models are valuable in drug screenings, disease modelling and personalised medicine, and can be reprogrammed and derived from somatic cells (e.g., skin fibroblasts, keratinocytes, peripheral blood mononuclear cells) from any patient or donor, undergoing differentiation protocols established in vitro (Takahashi and Yamanaka, 2006). They can subsequently be differentiated into several different lineages and cell types in considerable quantities, including cell types that are not accessible for experimental manipulation directly from individuals, such as cardiomyocytes (Devalla and Passier, 2018). Deriving cardiomyocytes from hiPSCs (hiPSC-CMs) can be efficiently done through defined small molecule differentiation protocols using chemically defined serum-free media (Lian et al, 2015; Bedada et al, 2016; Burridge et al, 2014; Mummery et al, 2012). More recently, differentiation protocols have been successfully adapted based on the molecular and genetic mechanisms that influence cardiomyocyte proliferation and differentiation throughout early cardiac development with the ultimate goal to generate cardiomyocytes in a high quantity in a reproducible manner (Maas et al, 2021; Buikema et al, 2020). In addition, these differentiation protocols can be carried out in both 2D and 3D settings, most commonly as monolayer cultures or embryoid bodies, respectively (Fig. 2) (Mummery et al, 2012).

2D hiPSC-CM culture systems have been widely used in the context of disease modelling and drug testing to obtain readouts on cardiac functional properties (i.e. electrophysiological parameters, calcium handling, and contractility), specific biochemical pathways, and cardiotoxicity at a cellular level (Shaheen et al, 2017; Abdelsayed et al, 2022; Dinani et al, 2023; Gilchrist et al, 2015). 2D culture systems are relatively easy to handle and analyse, although they cannot mimic the native tissue dynamics(Branco et al, 2019; Correia et al, 2018).

3D hiPSC-CM culture systems have emerged as a useful alternative that aims to provide a more complex and mature tissue platform since it can better mimic the cardiac microenvironment. There are two different strategies for the development of a 3D cardiac model, namely scaffold-free and scaffold-based techniques. In a scaffold-based approach focused on the extracellular-matrix (ECM)-cell interaction, the model can allow for the study of the interaction between external factors and cardiac mechanical function (Körner et al, 2021). Engineered heart tissues (EHTs) have been commonly used to establish novel disease models and drug screening platforms, allowing for control of mechanical

properties, electrical conductivity, and cell alignment (Breckwoldt et al, 2017; Eder et al, 2016). In this method, the constant mechanical strain aligns cells along force lines (Eder et al, 2016). In scaffold-free platforms, the model relies on cellular self-assembly and organisation in the absence of external cues, where cell–cell interactions play a major role. Cardiac organoids can be defined as hiPSC models in which cardiac cell-types self-organise during differentiation in vitro and spatially resemble cardiogenesis events, preserving the functional tissue architecture in long-term culture (Eglen and Reisine, 2019; Huebsch et al, 2016; Hofbauer et al, 2021; Drakhlis et al, 2021). Cardiac spheroids refer to co-culture models using a defined cell-type ratio that can be obtained using the hanging-drop technique, generating the microtissues from gravity formation of small droplets (Sharma and Gentile, 2021). They have been successfully developed including multiple cardiac cell types in order to more closely mimic the native tissue with vascularisation. These multi-cellular models have been applied in systems to study fibrosis in vitro (Figtree et al, 2017) and, when cultured in defined ratios, they have facilitated the investigation of the biochemical interactions between cardiomyocytes, fibroblasts or endothelial cells (Burridge et al, 2014).

Technical limitations need to be considered during hiPSC experimental design. Important model features to consider are accessibility, robustness, reproducibility, experimental manipulation, expertise required, throughput, scalability, automation and cost-effectiveness (Novakovic et al, 2014). Ultimately, the models' applications and readouts will dictate which of the above should take priority during model design (Abdelsayed et al, 2022). Developing 2D hiPSC-CMs high-throughput screening platforms for testing cardiotoxicity effects can be very effective in minimising the costs for discovery and experimentation of novel drug targets. On the other hand, 3D models allow to study of pathological mechanisms related with contractile force, and these have been recently adapted to high-throughput systems as well (Goßmann et al, 2020; Ma et al, 2023). Additionally, long-term assays are possible in 3D, facilitating the evaluation of long-term drug effects. Besides this, an important technical limitation of using hiPSC-CMs as a cardiac disease model is their immaturity resembling foetal-like cardiomyocytes (van den Berg et al, 2015). Key cardiomyocyte immaturity features have been noted when comparing hiPSC-CMs to adult cardiomyocytes, concerning cellular morphology (smaller cell size), electrophysiology (spontaneous beating), calcium handling (lower expression of calcium-handling proteins), contractility (less contraction force and sarcomere organisation), and metabolism (reduced mitochondria number and metabolic preference for glycolysis, instead of FAO) (Karbassi et al, 2020; Vučković et al, 2022). Therefore, more studies have centred on trying to surpass this challenge, and strategies have been tested and succeeded on improving hiPSC-CMs maturity such as 2D culture using substrates to resemble the stiffness of native tissue matrix (Jimenez-Vazquez et al, 2022), transition to a 3D culture format (Branco et al, 2019; Correia et al, 2018; Fong et al, 2016; Ulmer et al, 2018), co-culture of cardiomyocytes with other cell types (Giacomelli et al, 2020; Beauchamp et al, 2020; Varzideh et al, 2019; Napiwocki et al, 2021), long-term culture (Ergir et al, 2022), and exposure to adult-like metabolic substrates (fatty acids, oxidative substrates and low glucose) (Horikoshi et al, 2019; Lin et al, 2017; Feyen et al, 2020). The application of maturation strategies is considered during hiPSC experimental design based on study goals,

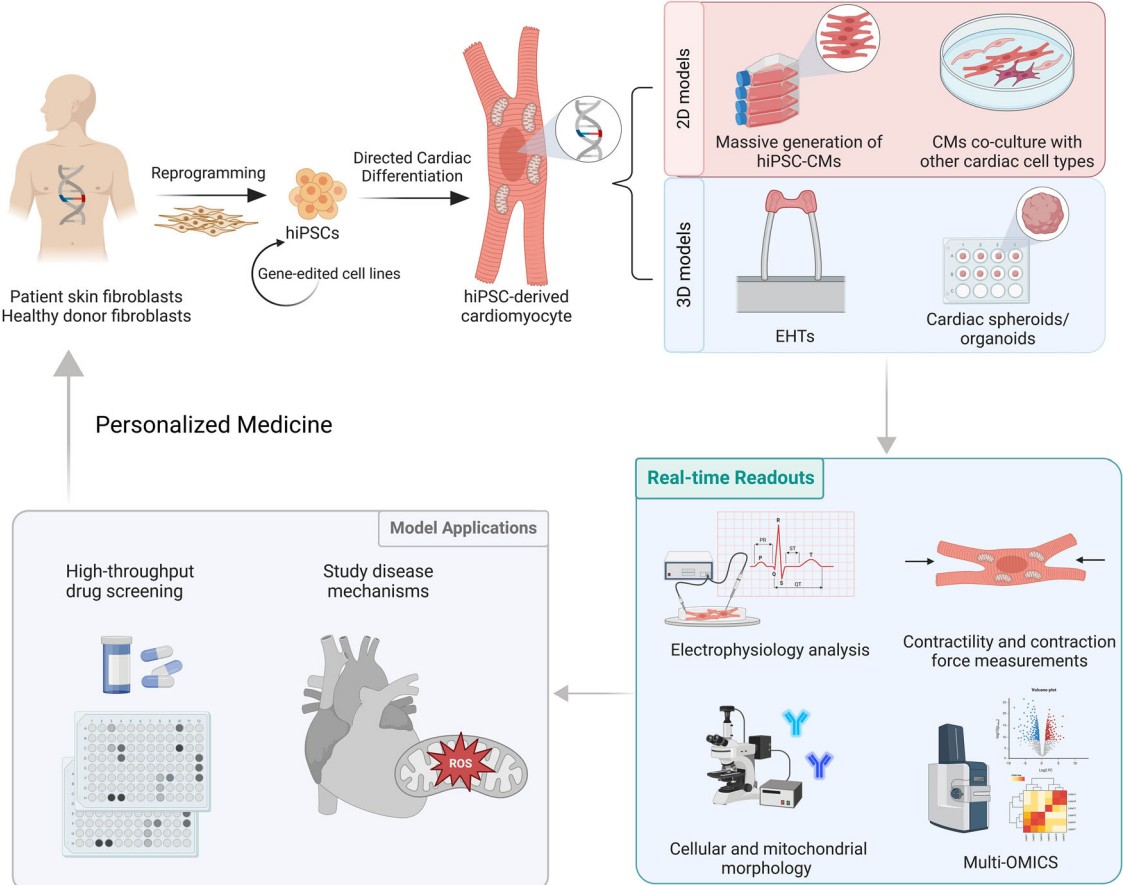

**Figure 2. Potential of hiPSC technology and models generation to personalise therapeutic options for metabolic cardiomyopathy patients.**

hiPSCs can be derived from somatic patient and healthy donor cells and differentiated into hiPSC-CMs. Relevant culture systems can be established based on research questions to be addressed with the possibility for 3D culture or co-culture with multiple cell types. Electrophysiology and contractility can be evaluated in tandem with mitochondrial function and metabolic activity, which can clarify currently unknown pathological mechanisms. Functional analysis of the model upon exposure to distinct pharmacological agents allows for an evaluation of patient-specific drug response. This can accelerate clinical decision-making, which is beneficial for the heterogeneous population of metabolic cardiomyopathy patients. Graphics were created with BioRender.com.

potentially excluding the need to achieve the adult cardiac phenotype if the target disease mechanism or drug response is identifiable at the foetal level.

Besides the maturation level of the model, cellular purity is also an important parameter when establishing a model to focus on cardiomyocyte functionality or drug response (Ban et al, 2017). During cardiomyocyte differentiation from hiPSC, a few strategies can be adopted to improve the culture purity and select only cardiomyocytes from undifferentiated and proliferative non-cardiac cells. This can include fibroblasts or undifferentiated hiPSC. Metabolic selection using glucose-depleted and lactate-supplemented medium is the most widely used technique since it is scalable, effective and non-genetic (Tohyama et al, 2013). Other alternatives include mRNA-based method (Wile et al, 2014), microRNA-based enrichment (Miki et al, 2015), and magnetic-activated cell sorting. Lastly, depending on the study aim, the disease model might need to be refined to include multiple cell lineages in the same system to account for cell–cell and systemic interactions (Sharma et al, 2020). Using hiPSC technologies combined with organ-on-chip approaches, it is possible to establish in 2D and 3D platforms and co-culture models using cardiomyocytes,

fibroblasts, and endothelial cells have proven to enhance cardiomyocyte maturation and more closely mimic the myocardium (Beauchamp et al, 2020; Varzideh et al, 2019; Napiwocki et al, 2021). hiPSC-derived cardiomyocytes have also been co-cultured with macrophages in order to assess their regenerative capacity in this setup, and functionality improvements were identified when this co-culture was established (Hitscherich et al, 2019; Xiao et al, 2025).

After hiPSC-CMs are generated and the model is established, a broad array of biochemical and functional assessments can be carried out using innovative technologies (Fig. 2). It is possible to characterise electrophysiological parameters (Crestani et al, 2020; Zhang et al, 2021), contractility efficiency (Wang et al, 2014), cardiomyocyte and mitochondrial morphology (Maria Cherian et al, 2023; Cao et al, 2020; Rohani et al, 2020; Yang et al, 2022) and obtain multi-omics datasets (Engel et al, 2023; Moore et al, 2023). These cell-based assays have been applied to hiPSC-CMs models of metabolic cardiomyopathies and contributed to a better understanding of energy metabolism perturbations and their direct consequences on cardiac structural and functional remodelling (Table 2).

# Metabolic cardiomyopathy modelling and therapy testing

A large number of hiPSC-CM studies on common inherited structural cardiomyopathies have showcased this technology's potential to decipher pathological mechanisms on a patient-specific basis (Ulmer and Eschenhagen, 2020). For instance, 2D hiPSC-CM models of HCM-associated variants have demonstrated that the cardiomyocytes' metabolic state is closely linked to their functionality (Mosqueira et al, 2018; Vučković et al, 2022). Along the same line, DCM-associated R9C *PLN* variant was linked to a metabolic shift in hiPSC-CMs from aerobic to anaerobic metabolism (Ceholski et al, 2018), and small molecule kinase inhibitors were able to rescue the DCM phenotype in hiPSC-CM, thereby improving contractility and metabolic function (Perea-Gil et al, 2022).

## hiPSC-derived cardiac models for lcFAODs

hiPSC-CMs models have recently been used to study lcFAODs, specifically VLCADD and MTPD (Chanana et al, 2016). To evaluate VLCADD disease mechanisms in relevant cells, fibroblasts from two VLCADD patients were reprogrammed into hiPSCs and differentiated into hiPSC-CMs (Knottnerus et al, 2020a). Electrophysiological irregularities were identified including shortening of action potentials, a significant increase in delayed afterdepolarisations (DADs), which are known to trigger arrhythmias, and elevated systolic and diastolic calcium concentrations. These were partly associated with the cardiac arrhythmic phenotype in patients with both the mild and severe phenotypes. The accumulation of FAO intermediates such as long-chain acylcarnitines was proposed as a possible disease mechanism linked to the VLCADD arrhythmogenic phenotype. Indeed, when VLCADD hiPSC-CMs were incubated with etomoxir, lowering the uptake of long-chain fatty acids by inhibiting CPT1, a considerable rescue of the electrophysiological abnormalities was seen in both mild and severe patient cell lines. Resveratrol also improved the long-chain acylcarnitines accumulation and rescued the cellular arrhythmia phenotype only in hiPSC-CMs from the patient with a mild phenotype (Knottnerus et al, 2020a). This was in line with previous results in skin fibroblasts, where resveratrol increased the lower amount of VLCAD with residual activity (Bastin et al, 2011). A third treatment that was tested in VLCADD hiPSC-CMs involved carnitine supplementation. Carnitine is often, preemptively, provided to patients with lcFAOD given that they accumulate acylcarnitines which can lead to a free carnitine deficiency, although this approach is contested (Vockley et al, 2020). Carnitine supplementation in hiPSC-CMs derived from a VLCADD patient with a mild phenotype did not improve the electrophysiology abnormalities, including DADs and action potential irregularities (Verkerk et al, 2021), suggesting a lack of benefit of this treatment, at least on cardiac electrophysiology.

A study on MTPD that used a CRISPR/Cas9-edited control hiPSC line to induce HADHA variants showed abnormal calcium handling, elongated action potentials, and abnormal repolarisation in MTPD hiPSC-CMs. Interestingly, when the MTPD hiPSC-CMs were exposed to fatty acids, structural abnormalities were detected, including sarcomere dissolution and mitochondrial damage. The mitochondria in MTPD hiPSC-CMs were smaller and swollen, presented poor cristae morphology and loss of mitochondrial

potential gradient when compared to control cells. In an attempt to reduce mitochondrial depolarisation, which was hypothesised to be causing the mitochondria's incapacity, SS-31 was used as treatment, which indeed fully rescued the proton leak in fatty acid-challenged MTPD hiPSC-CMs (Miklas et al, 2019). Moreover, a combination of microRNAs were evaluated in their potency to enhance hiPSC-CMs maturation through functional assays such as the measurement of cell area, force of contraction, metabolic capacity and electrophysiology. This resulted in a cocktail of miRs with relevant impact on the maturation levels.

## hiPSC-derived cardiac models for BTHS

Patient-derived hiPSC with distinct missense and splice site TAFAZZIN variants were used to create a human BTHS model from patient cells with severe and mild phenotypes. hiPSC were studied in an undifferentiated state, and disease-related features were identified, such as decreased basal oxygen consumption rates, reduced mitochondrial membrane potential, increased immature CL and higher levels of mitochondrial ROS. Moreover, these hiPSC lines were differentiated into cardiomyocytes and studied along with cardiomyocytes from Tafazzin-KD mice to investigate the BTHS cardiac-specific phenotype. Pathophysiological characteristics such as sarcomere disorganisation and structural remodelling of the respiratory chain complexes were identified in BTHS hiPSC-CMs (Dudek et al, 2016). Using one of the severe patient cell lines previously established (Dudek et al, 2013), the mitochondrial uptake of fatty acids was found to be decreased when the hiPSC-CMs were supplied with medium-chain FA dodecanoate covalently bound to a fluorescent dye (Kutschka et al, 2023).

Another group used unrelated patient-derived hiPSC-CMs with confirmed frameshift and missense TAFAZZIN variants, and also TAFAZZIN-deficient CRISPR/Cas9-edited hiPSC-CMs. These lines were cultured in 2D micropatterned fibronectin rectangles and in a 3D "heart-on-a-chip" system. BTHS hiPSC-CMs mitochondria were fragmented, and basal oxygen consumption rates elevated. Cardiomyocytes morphology was also analysed, and sarcomere assembly was impaired in the BTHS model in comparison to control cardiomyocytes. However, not all BTHS patient hiPSC-CMs presented a significantly different sarcomere organisation compared to control, highlighting the phenotypic heterogeneity and unclear genotype-phenotype correlations associated with BTHS. In addition, the BTHS hiPSC-CMs sarcomere impairment was not rescued when hiPSC-CMs were cultured in glucose-supplemented media. To study the BTHS contractility defects in a scaffold-based approach, hiPSC-CMs were cultured on muscular thin films, and this "heart-on-a-chip" was used to measure contractility. Sarcomere assembly was also impaired in BTHS hiPSC-CMs tissues, and contraction force was significantly weaker and not rescued when the tissues were supplied with glucose. All these findings showed that a severe cardiac phenotype in BTHS was partly recapitulated when cardiomyocyte morphology and function were analysed in 2D and 3D. Moreover, when glucose was provided and ATP levels restored, contractile dysfunction and sarcomere assembly were not rescued. In contrast, when the mutated TAFAZZIN was replaced in hiPSC-CMs by modified RNA transfection of the wild-type TAFAZZIN, both sarcomere misalignment and defective force generation were reversed to control levels. This indicated that the severe BTHS phenotype can be independent of myocardial energetic storage, and dependent on tafazzin deficiency (Wang et al, 2014).

Notably, this report provided strong support for a human clinical trial with SS-31 for patients with primary mitochondrial myopathy (Karaa et al, 2018). It is worth noting that, inherent to hiPSC pluripotency, other cell types can be generated in considerable amounts besides cardiomyocytes. This is particularly relevant for IMDs given their wide range of symptoms affecting distinct organs. A CPT2D model has been established using hiPSC-derived skeletal muscle myocytes (Yasuno et al, 2014) and LCHADD models focused on hiPSC-retinal pigment epithelial (RPE) cells to study LCHADD retinopathy (DeVine et al, 2021; Polinati et al, 2015). hiPSC-RPE cells derived from MELAS patients were used to study the link between impaired autophagy and mitochondrial recycling and function in the context of aging and macular degeneration (Bhattacharya et al, 2022). Focusing on neuronal MELAS pathogenesis, patient-derived hiPSC-neurons were used to study MELAS-associated mitophagy at a tissue-specific level (Hämäläinen et al, 2013). Patient-derived hiPSC lines were also generated for MMUT deficiency and differentiated into neurons, successfully establishing a human neuronal model of methylmalonic aciduria (Denley et al, 2025). A human neuronal model of Pompe disease was also developed from patient-derived hiPSCs and facilitated the identification of three compounds that enhanced GAA activity in neurons (Huang et al, 2019).

Ultimately, interesting cardiac pathophysiological insights have emerged from robust hiPSC studies. In Table 2, we present a summary of studies performed in mice models and cell-based platforms highlights how hiPSC systems have enabled the investigation of patient-specific cardiac disease mechanisms. Metabolic dysfunction can be further explored in metabolic cardiomyopathy models where it leads to cardiac structural abnormalities and, this way, new therapy alternatives can be suggested to common cardiomyopathies such as HCM (Ritterhoff and Tian, 2023) and DCM (Spoladore et al, 2023).

## Conclusions

IMDs can have a rapid and unpredictable onset, and genetic metabolic defects are the cue for highly heterogeneous cardiac phenotypes and symptoms. Deeper investigations into metabolic remodelling in cardiomyopathies are warranted to not only understand if this precedes disease onset but also how it contributes to disease progression. While animal models have contributed to understanding cardiac pathological mechanisms, it is clear that more physiologically relevant and patient-specific models are necessary to dissect disease mechanisms and predict drug responses. Of note, animal models are still regarded as essential to test the pharmacological activity and acute toxicity of new drugs prior to human clinical trials. However, 89% of novel therapy agents have failed in human clinical trials, where unpredicted human toxicity accounted for half of these failures (Van Norman, 2019). This way, hiPSCs-based models present a solution to accurately model diseases and determine tissue and patient-specific drug responses. This is possible with hiPSCs derived from patients' somatic cells that carry the genetic background of the patient and can be properly adapted to show sufficient maturity (Doss and Sachinidis, 2019). They can be a flexible tool used for drug discovery and for pre-clinical validation of repurposed drug combinations, having already potentiated the design of human clinical trials with beneficial outcomes (Karaa et al, 2018). In

lcFAODs and BTHS, hiPSC models have demonstrated their potential in reflecting patient heterogeneity and guiding personalised therapies (Table 2).

Animal models are currently highly debated, and limiting the use of animals in research is a high priority, not only by researchers but also other stakeholders such as patients and regulatory bodies (Stewart et al, 2023). Generally, the translation of basic and clinical research to the clinics requires numerous considerations regarding the optimal experimental design and model, where each should be considered according to the research questions and model applications149. Combining hiPSC and animal models may provide a more comprehensive experimental framework to investigate pathological mechanisms in more detail, enhancing translational success in personalised medicine. hiPSC models should be included as they can cover several tissue-specific phenotypes when differentiated into the cell types of interest, and this is especially relevant for IMDs with a wide range of organs affected.

## Pending issues

i).   Some IMD animal models are not viable due to high mortality rates, highlighting the need for further research into whether hiPSC-derived cardiac models can be a good alternative to predict long-term cardiac pathology in IMDs.

ii).  Despite the promise of hiPSC technologies, their high costs and technical limitations remain challenging to personalised medicine applications given the lack of standardisation in differentiation and functional maturation protocols for example. More efforts should be made to enhance the reproducibility and validation of hiPSC models.

iii). Future IMD studies of the cardiac phenotype should refine the experimental design to include interdependent research models to facilitate the transition from bench to bedside, and support a more sustainable model for drug development for these highly heterogeneous patient populations.

## Peer review information

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

## Acknowledgements

This work is financially supported by Human Measurement Models 2.0: for health research on disease and prevention, from NWO (no. 18953). Work in the Houtkooper group is supported by a grant from the Barth Syndrome Foundation. SM was supported by an International Postdoc grant from Independent Research Fund Denmark (1057-00039B) and by the Ministry of VWS Sectorplan 'Versnellen op Gezondheid'.

## Author contributions

**Adriana S Passadouro**: Conceptualisation; Visualisation; Writing—original draft; Writing—review and editing. **Berith M Balfoort**: Writing—original draft; Writing—review and editing. **Mirjam Langeveld**: Writing—original draft; Writing—review and editing. **Clara D M van Karnebeek**: Writing—review and editing. **Jolanda van der Velden**: Conceptualisation; Supervision; Funding acquisition; Writing—review and editing. **Riekelt H Houtkooper**: Conceptualisation; Supervision; Funding acquisition; Writing—review and editing. **Signe Mosegaard**: Conceptualisation; Supervision; Visualisation; Writing—original draft; Writing—review and editing.

## Disclosure and competing interest statement

The authors declare no competing interests.

