## [Peer Review File · EMBO Molecular Medicine]

Metabolic cardiomyopathies: untangling clinical heterogeneity with human stem-cell derived models

Adriana Passadouro, Berith Balfourt, Mirjam Langeveld, Clara van Karnebeek, Jolanda van der Velden, Riekelt Houtkooper, and Signe Mosegaard

Corresponding author(s): Signe Mosegaard (s.m.nielsen@amsterdamumc.nl) , Riekelt Houtkooper (r.h.houtkooper@amsterdamumc.nl)

Review Timeline:

Submission Date:	10th Apr 25
Editorial Decision:	28th Apr 25
Revision Received:	8th Aug 25
Editorial Decision:	2nd Sep 25
Revision Received:	5th Sep 25
Accepted:	9th Sep 25

Editor: Zeljko Durdevic

Transaction Report:

28th Apr 2025

Dear Dr. Mosegaard,

Thank you for the submission of your manuscript to EMBO Molecular Medicine. We have now received feedback from the two reviewers who agreed to evaluate your manuscript. As you will see from the reports below, the referees are positive about its interest and timeliness. However, referee #1 raises an important point about the narrow focus of the review. We agree with referee #1 and as it stands the manuscript appears to be better suited to a more specialized venue. We would however consider the manuscript for publication if you expand the discussion on other metabolic cardiomyopathies. Therefore, we would like to invite major revision to broaden the scope of the review to other metabolic disorders with cardiac manifestations. Further consideration of a revision that addresses reviewers' concerns in full will entail a second round of review.

I would also like to ask you to add the following items to your revised article:

- 1) Glossary: The glossary is meant to explain some of the terms used for laymen. Could you please identify terms that may need an "explanation"?
- 2) Pending issues: At the end of each article is a box highlighting issues that still need further studies and where research efforts should converge. Could you identify some pending issues?
- 3) Author contributions: Please remove it from the manuscript and specify author contributions in our submission system. CRediT has replaced the traditional author contributions section because it offers a systematic machine-readable author contributions format that allows for more effective research assessment. You are encouraged to use the free text boxes beneath each contributing author's name to add specific details on the author's contribution. More information is available in our guide to authors:
<https://www.embopress.org/page/journal/17574684/authorguide#authorshipguidelines>
- 4) If BioRender was used to create the figures, please add following sentence to the figure legends: "Graphics were created with BioRender.com."

I hope that the referees' comments do not prove too problematic to address and I look forward to reading your next version.

Yours sincerely,

Zeljko Durdevic

*** IMPORTANT INFORMATION ***

- 1) a .doc formatted version of the manuscript text (including Figure legends and tables)
- 2) Separate figure files
- 3) a letter INCLUDING the reviewer's reports and your detailed responses to their comments.

Also, and to save some time should your paper be accepted, please read below for additional information regarding some features of our research articles:

- 1) Glossary: EMBO Molecular Medicine articles will be accompanied by a glossary explaining some of the terms used for laymen. I identified the following:

_____, _____, _____

Could you please help us in identifying terms that may need an "explanation" other terms that we can add to the glossary.

2) Pending issues: At the end of each article we will have a box highlighting issues that still need further studies and where research efforts should converge (we call this the Pending issues box). From my reading I would say:

but I can see there may be many more. Could you work on this as well?

3) Disclosure and competing interest statement: Please include a statement declaring any competing commercial interests in relation to your submitted work.

4) Please note that we now mandate that all corresponding authors list an ORCID digital identifier. This takes <90 seconds to complete. We encourage all authors to supply an ORCID identifier, which will be linked to their name for unambiguous name identification.

Currently, our records indicate that the ORCID for your account is 0000-0002-4561-1978.

Link Not Available

-

Thank you,

Zeljko Durdevic

***** Reviewer's comments *****

Referee #1 (Remarks for Author):

Thank you for inviting me to read this interesting review by Passadouro et al. It contains a very nice summary of the metabolic and clinical features of long chain fatty acid oxidation disorders and Barth syndrome, as well as the current approaches to treat and experimentally model them. Particularly engaging was the description of the different manifestations of these disorders and the insights obtained from mouse and iPSC-derived cardiomyocytes.

Overall it is a fairly good balance of disease and model description, and could fit to this journal depending on the desired topic. For example, although the title and abstract heavily emphasize cardiomyopathies and human stem-cell derived models, the majority of the review is actually dedicated to describing fatty acid oxidation disorders and Barth syndrome. Further, nearly as much space is dedicated to describing animal models as human iPSC models in the text. For fatty acid oxidation disorders in particular, there is not a clear emphasis on cardiomyopathy in all sections. It does read as a nice review of these two interesting metabolic disorders, but I'm not sure if that is the point.

I would suggest the authors re-evaluate the goal of their review. If, as the title suggests, it is really about untangling metabolic cardiomyopathies using human stem-cell derived models, then all other aspects (i.e. other symptoms, treatments, animal models) need to be strongly de-emphasized, and additional metabolic disorders need to be discussed (even if these two are provided as proof-of-principle disorders). There are many other metabolic disorders with prominent cardiac manifestations (e.g. primary mitochondrial disorders, propionic aciduria), but these are completely ignored in this review. In particular, within the description of cardiomyocyte-derived stem cell models, the few papers the authors discuss are interesting, but further literature would be required to make broad conclusions about the suitability and limitations of these cells to model cardiomyopathies in metabolic disorders.

If instead, the authors mean to provide an interesting review of fatty acid oxidation disorders and Barth syndrome, with a small highlight on the role of cardiac symptoms and the potential of stem cell derived cardiomyocytes to bring new insights, then they only need to change the title and abstract to more accurately reflect the current manuscript text.

Other:

- Abstract: The authors should remove the final phrase "...including high-throughput drug screening for personalized therapeutic interventions." as this is actually not discussed in the manuscript.
- pg 10-11: Please use the correct protein/cDNA nomenclature (e.g. p.Met40Lys, c.1528G>C) and indicate whether the described variants are numbered according to mouse or human.
- pg 20: The paragraph starting with "Notably, genome editing..." does not fit to the topic being discussed in this section, and can/should be removed
- pg 21: The paragraph starting with "We will discuss the studies..." is unnecessary and can/should be removed.
- Table 1: This is very useful. But are all clinical feature listed actually present in both fatty acid oxidation disorders and Barth Syndrome? If so, this should be indicated somewhere. If not, extra columns should be provided to indicate which symptoms are present in each disorder.
- Figure 1 & 2: These figures could be merged. Most of what is indicated in Figure 2 is already displayed in the mitochondrial schematic in the top left of Figure 1. Outside of this mitochondria, the additional information of Figure 1 is not very clear or easy to see, and takes up a lot of space. I would suggest either removing them or combining the information of the right hand side and middle of Figure 1 with that presented in the bottom left of Figure 1 (bottom left and bottom right are replicated anyway).
- Figure 3: This figure is not really discussed in the text, and on is not very useful. Better would be to provide a figure depicting the different approaches to 2D and 3D model cardiomyocytes, which is currently discussed extensively in the text.

Referee #2 (Remarks for Author):

In this nice review, Passadouro and others summarize current understanding of metabolic cardiomyopathies. Namely, they focus on the pathology of long chain fatty acid oxidation disorders along with Barth Syndrome. The aim of the review is to highlight the challenges associated with current animal models and highlight the utility of more relevant (albeit in vitro) human preclinical models including hiPSCs.

Overall, the summary is quite thorough and the points raised regarding the inability of current preclinical models to predict therapeutic efficacy of drug candidates is compelling. I have a few minor comments that I hope will be helpful to the authors.

1. Introduction, second paragraph, third line: should read "At the cardiomyocyte level".
2. Page 20, third paragraph, line 8; typo "to to".
3. Page 21, last paragraph: It is also worthy to note that the challenge associated with cell purity (i.e. what percentage of hiPSCs-CMs are actually CMs and not some undifferentiated contaminant cell).
4. Page 26, it is important to note the challenge inherent with in vitro human disease models including the reductionist approach that excludes the roles of other critical cell types including endothelial, fibroblast, macrophages, and others and efforts that have been made to circumvent these challenges.

Dear Dr. Durdevic, dear Zeljko

We thank you for considering our manuscript with major revisions and hereby send you updated manuscript. We have revised the manuscript according to the reviewers' constructive suggestions, and included a "Pending Issues" section (line 785-796). We also included a Glossary section just after the abstract. Please find a point-by-point response to the reviewers below

Reviewer #1:

Thank you for inviting me to read this interesting review by Passadouro et al. It contains a very nice summary of the metabolic and clinical features of long chain fatty acid oxidation disorders and Barth syndrome, as well as the current approaches to treat and experimentally model them. Particularly engaging was the description of the different manifestations of these disorders and the insights obtained from mouse and iPSC-derived cardiomyocytes.

Overall it is a fairly good balance of disease and model description, and could fit to this journal depending on the desired topic. For example, although the title and abstract heavily emphasize cardiomyopathies and human stem-cell derived models, the majority of the review is actually dedicated to describing fatty acid oxidation disorders and Barth syndrome. Further, nearly as much space is dedicated to describing animal models as human iPSC models in the text. For fatty acid oxidation disorders in particular, there is not a clear emphasis on cardiomyopathy in all sections. It does read as a nice review of these two interesting metabolic disorders, but I'm not sure if that is the point.

I would suggest the authors re-evaluate the goal of their review. If, as the title suggests, it is really about untangling metabolic cardiomyopathies using human stem-cell derived models, then all other aspects (i.e. other symptoms, treatments, animal models) need to be strongly de-emphasized, and additional metabolic disorders need to be discussed (even if these two are provided as proof-of-principle disorders). There are many other metabolic disorders with prominent cardiac manifestations (e.g. primary mitochondrial disorders, propionic aciduria), but these are completely ignored in this review.

We appreciate the interest and constructive feedback on our manuscript. We agree that it would enhance the quality of this review if a broader discussion of IMDs that include cardiac presentations is included. To include this in the manuscript we have expanded the section entitled "Clinical heterogeneity in metabolic cardiomyopathies" on page 6 to 8. Moreover, we added a more detailed overview of IMDs with well-established cardiac phenotypes, also including their prevalence, the affected gene, and their non-cardiac phenotypes (Table 1; page 45-48). Additionally, we provide a comprehensive summary in supplementary table 1 of all IMDs that have reported cardiac manifestations. This data has been extracted from the established and highly curated IEM database (www.IEMbase.org) using the search term "cardiomyopathy". However, since some of these diseases do not typically manifest with cardiomyopathy, we decided to add the list as a supplementary table.

In particular, within the description of cardiomyocyte-derived stem cell models, the few papers the authors discuss are interesting, but further literature would be

required to make broad conclusions about the suitability and limitations of these cells to model cardiomyopathies in metabolic disorders.

We extended the table on cardiac models (*in vivo* and hiPSC-based) on more IMDs such as MELAS, Pompe, Danon and Fabry diseases (Table 1; page 45-48). In the text, we also showcased how hiPSC models have contributed to understanding the cardiac phenotype of more common structural cardiomyopathies such as HCM and DCM (line 656 to 662)

If instead, the authors mean to provide an interesting review of fatty acid oxidation disorders and Barth syndrome, with a small highlight on the role of cardiac symptoms and the potential of stem cell derived cardiomyocytes to bring new insights, then they only need to change the title and abstract to more accurately reflect the current manuscript text.

As mentioned earlier in our answer, we agree with the reviewer that it will be more relevant to expand the discussion. We therefore decided not to change the title and abstract but instead adapt and broaden the manuscript with the changes mentioned in the answer above.

Other:

-Abstract: The authors should remove the final phrase "...including high-throughput drug screening for personalized therapeutic interventions." as this is actually not discussed in the manuscript.

We fully agree and have removed the sentence from the abstract.

-pg 10-11: Please use the correct protein/cDNA nomenclature (e.g. p.Met40Lys, c.1528G>C) and indicate whether the described variants are numbered according to mouse or human.

We thank the reviewer for noticing this, we have added that this is a model reflecting the human c.1528G>C variant.

-pg 20: The paragraph starting with "Notably, genome editing..." does not fit to the topic being discussed in this section, and can/should be removed

We agree and have removed this paragraph.

-pg 21: The paragraph starting with "We will discuss the studies..." is unnecessary and can/should be removed.

We agree and have removed this paragraph.

-Table 1: This is very useful. But are all clinical feature listed actually present in both fatty acid oxidation disorders and Barth Syndrome? If so, this should be indicated somewhere. If not, extra columns should be provided to indicate which symptoms are present in each disorder.

This is now table 2. The clinical features of these IMDs are summarized in the manuscript text for IcFAODs from line 223 to 229 and for BTHS this description can be found from line 406 to 417. Summarized, the clinical findings can now also be found in table 1, page 45.

-Figure 1 & 2: These figures could be merged. Most of what is indicated in Figure 2 is already displayed in the mitochondrial schematic in the top left of Figure 1. Outside of this mitochondria, the additional information of Figure 1 is not very clear or easy to see, and takes up a lot of space. I would suggest either removing them or combining the information of the right hand side and middle of Figure 1 with that presented in the bottom left of Figure 1 (bottom left and bottom right are replicated anyway).

We agree with the reviewer that there are overlaps of the two figures, and indeed the information in Figure 2 is not in detail as relevant for this review. Based on the reviewers comment we decided to delete figure 2 from the manuscript.

-Figure 3: This figure is not really discussed in the text, and on is not very useful. Better would be to provide a figure depicting the different approaches to 2D and 3D model cardiomyocytes, which is currently discussed extensively in the text.

This is now figure 2, the figure is now discussed on several positions in the text, line 585 and more extensively in line 663 to 671.

Reviewer #2:

In this nice review, Passadouro and others summarize current understanding of metabolic cardiomyopathies. Namely, they focus on the pathology of long chain fatty acid oxidation disorders along with Barth Syndrome. The aim of the review is to highlight the challenges associated with current animal models and highlight the utility of more relevant (albeit in vitro) human preclinical models including hiPSCs.

Overall, the summary is quite thorough and the points raised regarding the inability of current preclinical models to predict therapeutic efficacy of drug candidates is compelling. I have a few minor comments that I hope will be helpful to the authors.

We appreciate the feedback from the reviewer and have included all changes suggested.

1. Introduction, second paragraph, third line: should read "At the cardiomyocyte level".

We thank the reviewer for this comment and have corrected the text.

2. Page 20, third paragraph, line 8; typo "to to".

This has been corrected.

3. Page 21, last paragraph: It is also worthy to note that the challenge associated with cell purity (i.e. what percentage of hiPSCs-CMs are actually CMs and not some undifferentiated contaminant cell).

This comment raised a very important note that was missing in our discussion but this can now be found from line 645 to 649.

4. Page 26, it is important to note the challenge inherent with in vitro human disease models including the reductionist approach that excludes the roles of other critical cell types including endothelial, fibroblast, macrophages, and others and efforts that have been made to circumvent these challenges.

We have added this subject and more literature on these efforts from line 649 to 656.

2nd Sep 2025

Dear Dr. Mosegaard,

Thank you for the submission of your manuscript to EMBO Molecular Medicine. I am pleased to inform you that we will be able to accept your manuscript pending the following final amendments:

- 1) Please implement referee's suggestions.
- 2) There is a callout for Table EV1. Please correct/update.
- 3) Rename "Conflicts of interests" to "Disclosure and competing interests statement". We updated our journal's competing interests policy in January 2022 and request authors to consider both actual and perceived competing interests. Please review the policy <https://www.embopress.org/competing-interests> and update your competing interests if necessary.
- 4) Please correct the reference citation in the reference list. Where there are more than 10 authors on a paper, 10 will be listed, followed by "et al.". Also, please remove DOIs. Please check "Author Guidelines" for more information. <https://www.embopress.org/page/journal/17574684/authorguide#referencesformat>
- 5) As part of the EMBO Publications transparent editorial process initiative EMBO Molecular Medicine will publish online a Review Process File (RPF) to accompany accepted manuscripts. This file will be published in conjunction with your paper and will include the anonymous referee reports, your point-by-point response and all pertinent correspondence relating to the manuscript. Let us know whether you agree with the publication of the RPF.

I look forward to receiving the revised version of your manuscript.

Yours sincerely,

Zeljko Durdevic

Zeljko Durdevic
Senior Editor
EMBO Molecular Medicine

*** IMPORTANT INFORMATION ***

- 1) a .doc formatted version of the manuscript text (including Figure legends and tables)
- 2) Separate figure files
- 3) a letter INCLUDING the reviewer's reports and your detailed responses to their comments.

Also, and to save some time should your paper be accepted, please read below for additional information regarding some features of our research articles:

- 1) Glossary: EMBO Molecular Medicine articles will be accompanied by a glossary explaining some of the terms used for laymen. I identified the following:

_____, _____, _____

Could you please help us in identifying terms that may need an "explanation" other terms that we can add to the glossary.

- 2) For more information: This is a short list of related web links for further consultation by the readers. Could you identify some

relevant ones? Examples are patient associations, OMIM related links, databases, authors websites, etc.

3) Pending issues: At the end of each article we will have a box highlighting issues that still need further studies and where research efforts should converge (we call this the Pending issues box). From my reading I would say:

but I can see there may be many more. Could you work on this as well?

4) Disclosure and competing interest statement: Please include a statement declaring any competing commercial interests in relation to your submitted work.

5) Please note that we now mandate that all corresponding authors list an ORCID digital identifier. This takes <90 seconds to complete. We encourage all authors to supply an ORCID identifier, which will be linked to their name for unambiguous name identification.

Currently, our records indicate that the ORCID for your account is 0000-0002-4561-1978.

Link Not Available

-

Thank you,

Zeljko Durdevic

***** Reviewer's comments *****

Referee #1 (Remarks for Author):

Thanks to the authors for their conscientious revision of this article. It now presents a broader, more balanced discussion of cardiomyopathy due to inherited metabolic disorders. In particular, Table 1 is quite valuable. Although I don't believe it is exhaustive, I agree with the approach of using IEMbase to define which disorders to include. There are still a few minor proofing aspects (e.g. "in this review we" is used multiple times but could be limited to the abstract, the abbreviation CPT2D is used before defining and describing this condition) which might enhance readability. Overall the authors are to be congratulated on their nice work.

All editorial and formatting issues were resolved by the authors.

9th Sep 2025

Dear Dr. Mosegaard,

We are pleased to inform you that your manuscript is accepted for publication and is now being sent to our publisher to be included in the next available issue of EMBO Molecular Medicine.

Your manuscript will be processed for publication by EMBO Press. It will be copy edited and you will receive page proofs prior to publication. You will soon be contacted by Springer Nature to sign your publishing license. When you login to the customer service website, please use the following token to waive the article publication charges. Should you experience any difficulty, please email publishing@embo.org.

Waiver token: *token unavailable*

Zeljko Durdevic
Senior Editor
EMBO Molecular Medicine